# Simplex-Aligned Diffusion with Cross-Granularity Interaction for Robust Medical Image Classification

**Chao Wu** iD                                                   CWU64@BUFFALO.EDU

**Mingchen Gao** iD                                              MGAO8@BUFFALO.EDU

*Department of Computer Science and Engineering, University at Buffalo (SUNY), Buffalo, NY, United States*

**Editors:** Accepted for publication at MIDL 2026

## Abstract

The clinical deployment of medical image classification systems hinges on their trustworthiness, specifically, the ability to provide calibrated uncertainty estimates and maintain robustness under acquisition shifts. While generative diffusion models offer promising distributional modeling, existing approaches suffer from a fundamental geometric conflict: they apply unbounded Gaussian noise directly to bounded label simplices. We identify that this theoretical mismatch forces predictions into invalid probability spaces, serving as a primary source of model unreliability and overconfidence. To resolve this, we propose Simplex-Aligned Diffusion. Unlike standard methods, we reformulate the label generation process on an unconstrained logit manifold. By mapping the probability simplex to a Euclidean space, we ensure mathematical consistency with Gaussian diffusion, which effectively acts as a geometric regularizer for uncertainty calibration. Furthermore, we introduce a Transformer-based Cross-Granularity Interaction module to stabilize visual guidance by dynamically modeling global-local dependencies. Extensive experiments on the APTOS2019 and HAM10000 benchmarks demonstrate that our framework not only achieves competitive accuracy but significantly outperforms state-of-the-art baselines in calibration error (ECE) and resilience to clinical artifacts (e.g., sensor noise, blur), offering a mathematically rigorous and clinically reliable paradigm.[1]

**Keywords:** Robust Medical Image Classification, Simplex-Aligned Diffusion, Uncertainty Calibration

## 1. Introduction

Medical image classification is a cornerstone of modern computer-aided diagnosis. While discriminative models like CNNs (He et al., 2016) and ViTs (Dosovitskiy, 2020) have achieved high benchmarks, they often exhibit fragility in clinical reality. Medical images are inherently plagued by noise and artifacts, and traditional models tend to exploit spurious correlations or "shortcuts" (Geirhos et al., 2020), leading to over-confident predictions even under significant distribution shifts (Esteva et al., 2019; Li et al., 2025a; Hu et al., 2026). Consequently, constructing a system that is not only accurate but also trustworthy, providing calibrated uncertainty estimates, and maintaining robustness, remains a critical challenge for patient safety (Sha et al., 2025).

To mitigate these limitations, the field has witnessed a shift towards Generative Classifiers leveraging Denoising Diffusion Probabilistic Models (DDPMs) (Ho et al., 2020). Current explorations primarily follow two paths. The first path reformulates classification as

---

1. Code is available at https://github.com/SamaritanW/simplex-aligned-diffusion

conditional image generation, comparing reconstruction error (MSE) to assign labels (Favero et al., 2025; Müller-Franzes et al., 2022). However, this strategy suffers from a critical inductive bias: pixel-level reconstruction quality does not equate to diagnostic correctness (Zhang et al., 2018), and the iterative sampling required for every class is computationally prohibitive for clinical workflows (Chen et al., 2023a).

The second path models classification directly as a "label generation" process. While efficient, existing frameworks (Yang et al., 2023, 2025; Han et al., 2022) face a fundamental reliability hazard stemming from a geometric conflict. They apply unbounded latent Gaussian noise to discrete One-Hot label vectors constrained to a bounded probability simplex. This theoretical mismatch forces noisy states off the valid manifold, hindering modeling precision and causing the model to learn a biased posterior that underestimates uncertainty (Hoogeboom et al., 2021; Austin et al., 2021). Furthermore, existing architectures (Shen et al., 2021; Yang et al., 2025; Li et al., 2025b) often rely on static fusion for conditional guidance, overlooking the dynamic semantic dependency between global anatomical contexts and local lesions.

To address these challenges, we propose a novel Simplex-Aligned Diffusion framework. We are the first to reformulate label generation from the discrete one-hot simplex to the continuous logit manifold for medical classification. This mapping acts as an intrinsic geometric safety constraint, ensuring mathematical consistency with Gaussian diffusion. The main contributions are:

- We propose a generative classification strategy that operates in the logit space. This approach effectively resolves the theoretical conflict between simplex constraints and Gaussian noise assumptions, providing a mathematically consistent solution for robust label generation.

- Through systematic evaluation under ImageNet-C style input corruptions(e.g., sensor noise, blur)[2], we reveal that while maintaining competitive accuracy on clean data, our logit-based diffusion demonstrates significantly superior resilience to artifacts and uncertainty calibration compared to standard one-hot diffusion baselines.

- We design a Transformer-based interaction module to refine the feature coupling within the frozen encoder. This mechanism explicitly models the dependency between global and local views, ensuring the diffusion model receives stable and precise visual guidance without requiring heavy architectural changes.

## 2. Related Works

**Diffusion Models for Medical Analysis.** DDPMs (Ho et al., 2020) have surpassed GANs in generative fidelity and are now widely adopted in medical image synthesis (Müller-Franzes et al., 2022; Khader et al., 2023). Beyond generation, diffusion models have been adapted for discriminative tasks. In segmentation, they model the joint distribution of images and masks to produce uncertainty-aware predictions (Amit et al., 2021; Wu et al., 2024,

---

2. We distinguish *input corruptions* (e.g., sensor noise, blur) used for robustness testing from the *latent generative noise* used within the diffusion process.

2026). In detection and anomaly localization, diffusion-based methods leverage reconstruction errors or latent-space signals to highlight pathological regions (Wolleb et al., 2022; Pinaya et al., 2022). These developments demonstrate the strong capability of diffusion models to capture complex structures inherent in medical data.

**Generative Classification Frameworks.** Current diffusion-based classifiers primarily follow two paradigms. The first treats classification as conditional generation, determining labels by evaluating the reconstruction likelihood of the input image under different class conditions (Favero et al., 2025; Li et al., 2023a). While explainable, this reconstruction-based approach is computationally expensive due to the need for repeated sampling per class. The second paradigm, pioneered by CARD (Han et al., 2022), formulates classification as a generative process of the label vector itself ($p(y|x)$). In the medical domain, DiffMIC (Yang et al., 2023) and DiffMIC-v2 (Yang et al., 2025) adopted this strategy, introducing dual-granularity guidance to improve feature extraction. However, these methods typically impose Gaussian noise directly on One-Hot encoded labels. As noted in recent theoretical discussions (Hoogeboom et al., 2021), applying continuous Gaussian diffusion to discrete or simplex-constrained data introduces a fundamental geometric mismatch, potentially limiting modeling precision.

**Diffusion on Constrained Manifolds.** Existing diffusion methods for discrete data rely on discrete transition kernels (Hoogeboom et al., 2021; Austin et al., 2021) or bit-to-real relaxations (Chen et al., 2023b). Yet, a geometrically coherent mapping from the probability simplex to a continuous logit space has not been explored in medical classification. We address this gap by introducing a simplex-consistent framework that preserves the robustness benefits of Gaussian diffusion.

## 3. Methodology

### 3.1. Problem Formulation: Simplex-Aligned Manifold

We formulate the classification task as a conditional generative process. Let $\mathcal{D} = \{(\mathbf{x}, \mathbf{y})\}$ be the dataset, where $\mathbf{y} \in \Delta^{C-1}$ is the one-hot label on the probability simplex.

**Geometric Conflict in One-Hot Diffusion.** Standard diffusion models define the forward process on $\mathbf{y}$ as a Gaussian transition:

$$\mathbf{y}_t = \sqrt{\bar{\alpha}_t}\mathbf{y} + \sqrt{1 - \bar{\alpha}_t}\boldsymbol{\epsilon}, \quad \boldsymbol{\epsilon} \sim \mathcal{N}(\mathbf{0}, \mathbf{I}). \tag{1}$$

Since $\mathbf{y}$ is bounded (sparse and non-negative) while $\boldsymbol{\epsilon}$ is unbounded, the noisy state $\mathbf{y}_t$ inevitably falls outside the valid simplex (i.e., $\mathbf{y}_t \notin \Delta^{C-1}$), rendering the geometric structure ill-defined during generation. We provide a mathematical proof in Appendix F, demonstrating that this mismatch introduces a systematic bias (Proposition 1) and renders standard training objectives intractable (Proposition 2).

**Intuition for the Geometric Conflict.** Intuitively, this support mismatch between unbounded Gaussian noise and the bounded probability simplex leads to two fundamental hazards (see detailed proofs in Appendix F):

- **Probability Leakage (Proposition 1):** Since Gaussian noise is defined on $\mathbb{R}^C$, the diffusion process inevitably pushes noisy states into "invalid" regions (e.g., values $< 0$

or $> 1$). This causes a systematic boundary bias where the model underestimates the values needed to reach the simplex vertices, leading to over-confident yet miscalibrated "over-smoothed" predictions.

- **Approximation Gap (Proposition 2):** Simply forcing states back onto the simplex (e.g., via Softmax) at each step breaks the linear superposition property of Gaussian diffusion. This renders the training objective a biased proxy (Jensen's Gap), preventing the model from converging to the true data manifold.

To resolve these, we propose performing diffusion on the unconstrained logit manifold $\mathbf{z}_0$, which is naturally compatible with Gaussian assumptions.

**Simplex-Aligned Logit Diffusion.** To resolve this, we propose performing diffusion on a diffeomorphic continuous manifold. We map $\mathbf{y}$ to a centered logit state $\mathbf{z}_0$ via a scaled Center Log-Ratio (CLR) transformation:

$$\mathbf{z}_0 = \mathcal{T}(\mathbf{y}) = \frac{1}{\lambda}\left(\log(\mathbf{y}_{\mathrm{smooth}}) - \frac{1}{C}\sum_{k=1}^{C}\log(y_{\mathrm{smooth}}^{(k)})\right). \tag{2}$$

Here, $\mathbf{z}_0 \in \mathbb{R}^C$ resides in an unconstrained Euclidean space compatible with Gaussian noise. The forward process $\mathbf{z}_t = \sqrt{\bar{\alpha}_t}\mathbf{z}_0 + \boldsymbol{\epsilon}$ is now mathematically consistent, preserving the manifold structure throughout the diffusion chain. We utilize label smoothing ($\mathbf{y}_{\mathrm{smooth}}$ with $\epsilon = 0.001$) to handle numerical singularities.

### 3.2. Transformer-Enhanced Visual Tokenizer

To provide context-aware guidance, we adopt the dual-stream feature extraction paradigm from DiffMIC-v2 (Yang et al., 2025) but significantly enhance the feature interaction mechanism using a Transformer encoder.

**Dual-Stream Extraction.** The Global Stream uses an encoder $E_g$ to extract a holistic token $\mathbf{f}_g$ and a spatial saliency map $\mathbf{S}$. Guided by $\mathbf{S}$, the Local Stream extracts $K$ discriminative patches and encodes them into regional tokens $\{\mathbf{f}_l^k\}_{k=1}^K$. We explicitly extract the global prior $\mathbf{v}_g$ and local prior $\mathbf{v}_l$ (logits) directly from these streams before interaction. These priors represent the initial predictions from global and local views, respectively.

**Cross-Granularity Interaction.** Existing methods often combine global and local features via simple concatenation or static fusion. To explicitly model the dynamic dependency between the holistic context and local nuances, we construct a unified sequence $\mathbf{Z}^{(0)} = [\mathbf{f}_g, \mathbf{f}_l^1, \ldots, \mathbf{f}_l^K]$. We feed this sequence into a Transformer Encoder Layer to perform deep interaction:

$$\mathbf{Z}' = \mathrm{LN}(\mathbf{Z}^{(0)} + \mathrm{MSA}(\mathbf{Z}^{(0)})), \qquad \mathbf{Z}^{(1)} = \mathrm{LN}(\mathbf{Z}' + \mathrm{FFN}(\mathbf{Z}')). \tag{3}$$

**Output Formulation.** The tokenizer yields two refined conditions for the diffusion process: 1) The first token of $\mathbf{Z}^{(1)}$ is projected to obtain the fusion prior $\mathbf{v}_{\mathrm{trans}}$ (used for loss weighting). 2) The subsequent tokens form the **Refined Semantic Features** $\mathbf{F}_{\mathrm{ref}} = \mathbf{Z}_{1:K+1}^{(1)}$, which serve as the high-level semantic condition injected into the UNet.

### 3.3. Generative Process with Refined Semantic Feature Injection

We model classification as a reverse diffusion process refining $\mathbf{z}_t$ to $\mathbf{z}_0$. First, following (Yang et al., 2025), we concatenate a spatial guidance map $\mathcal{M}$ (derived from $\mathbf{v}_g, \mathbf{v}_l$) with $\mathbf{z}_t$. Second, for semantic injection, we inject the Transformer-derived Refined Semantic Features $\mathbf{F}_{\mathrm{ref}}$ into the U-Net via an Adaptive Channel Gating mechanism. $\mathbf{F}_{\mathrm{ref}}$ is projected and fused with the intermediate feature map $\mathbf{h}$ of the U-Net to compute a channel-wise gating weight $\mathbf{w}$. The feature map is modulated as $\mathbf{h}' = \mathbf{h} \cdot \mathrm{Softmax}(\mathbf{w})$.

**Optimization.** The diffusion model $\boldsymbol{\epsilon}_\theta$ predicts the noise $\boldsymbol{\epsilon}$. The training objective is a re-weighted MSE loss:

$$\mathcal{L}_{\boldsymbol{\epsilon}} = \mathbb{E}_{t, \mathbf{z}_0, \boldsymbol{\epsilon}} \left[ \omega(\mathbf{v}_{\mathrm{trans}}) \cdot \|\boldsymbol{\epsilon} - \boldsymbol{\epsilon}_\theta(\mathbf{z}_t, t, \mathcal{M}, \mathbf{F}_{\mathrm{ref}})\|^2 \right], \tag{4}$$

where $\omega(\cdot)$ is a focal term derived from the Transformer's fusion prior $\mathbf{v}_{\mathrm{trans}}$, enforcing focus on hard samples.

**Reweighting of the Diffusion Loss.** In Eq. (4), the weighting function $\omega(\cdot)$ implements a focal-style sample reweighting mechanism applied directly to the diffusion noise regression objective. Unlike introducing an auxiliary classification loss, this term modulates the contribution of each training sample within the diffusion denoising loss itself, based on the model's confidence in the ground-truth class.

Specifically, let $\mathbf{p} = \mathrm{Softmax}(\mathbf{v}_{\mathrm{trans}})$ denote the class probability vector obtained from the Transformer fusion prior $\mathbf{v}_{\mathrm{trans}}$, and let $p_y$ represent the predicted probability assigned to the true class $y$. The weighting function is defined as

$$\omega(\mathbf{v}_{\mathrm{trans}}) = 1 + \alpha(1 - p_y)^\gamma, \tag{5}$$

where $\alpha$ and $\gamma$ are scalar hyperparameters controlling the overall strength and focusing behavior of the reweighting, respectively.

This formulation assigns larger weights to low-confidence (hard) samples while down-weighting high-confidence (easy) samples, while maintaining a minimum weight of 1. As a result, the diffusion model is encouraged to allocate more capacity to ambiguous or challenging cases without destabilizing training. Importantly, this focal-style weighting operates solely on the diffusion noise prediction error and does not introduce an additional discriminative objective. We therefore preserve the generative nature of the framework while improving optimization focus on difficult samples.

### 3.4. Inference Procedure

During inference, we initiate the process by sampling a random Gaussian noise $\mathbf{z}_T \sim \mathcal{N}(\mathbf{0}, \mathbf{I})$. We then iteratively denoise this state to recover the estimated logit vector $\hat{\mathbf{z}}_0$ via the reverse diffusion process. This generation is conditioned on the structural guidance map $\mathcal{M}$ and the refined semantic features $\mathbf{F}_{\mathrm{ref}}$, ensuring the generated logits are semantically consistent with the input image. Finally, the discrete class probability is recovered as $\hat{\mathbf{p}} = \mathrm{Softmax}(\lambda \cdot \hat{\mathbf{z}}_0)$, serving as the final classification prediction.

## 4. Experiments

We evaluate our method on two public benchmarks: **APTOS2019** (Karthik et al., 2019) (Retina) and **HAM10000** (Tschandl et al., 2018) (Dermatoscopy). Following standard

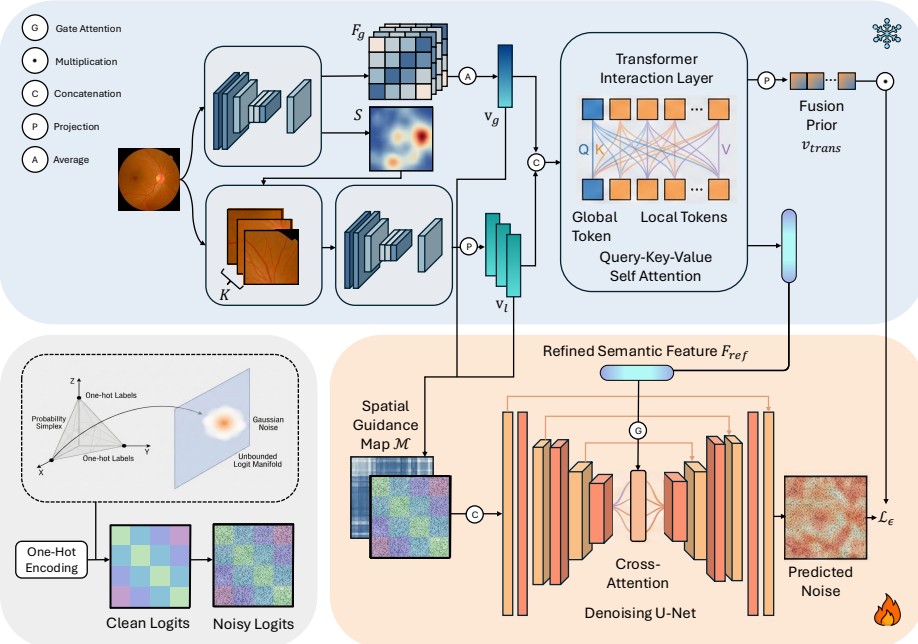

Figure 1: Overview of the proposed framework. The method builds upon a dual-stream backbone enhanced by two key innovations: (1) **Transformer-Enhanced Interaction**: Global and local features are fused via a Transformer layer to yield refined semantic features $\mathbf{F}_{ref}$ and a fusion prior. Explicit priors $\mathbf{v}_g, \mathbf{v}_l$ from the backbone are used to construct the spatial map. (2) **Simplex-Aligned Diffusion**: Unlike standard approaches operating on discrete one-hot vectors, our model operates on the continuous logit manifold $\mathbf{z}_0$, receiving structural guidance from $\mathcal{M}$ and semantic guidance from $\mathbf{F}_{ref}$ to iteratively denoise the target logits.

protocols (Yang et al., 2025), we employ Accuracy (Acc), Macro F1-score (F1), and Cohen's Kappa ($\kappa$) (Cohen, 1960) as evaluation metrics. Detailed dataset statistics, data splitting strategies, and mathematical definitions of these metrics are provided in Appendix A and Appendix E.

### 4.1. Main Results

Table 1 presents the quantitative comparison against state-of-the-art methods. We benchmark against specialized imbalanced learning strategies, including LDAM (Cao et al., 2019), OHEM (Shrivastava et al., 2016), MTL (Liao and Luo, 2017), DANIL (Gong et al., 2020), CL (Marrakchi et al., 2021), and ProCo (Yang et al., 2022), as well as advanced architectures such as DGCMM (Wang et al., 2022), UniFormer (Li et al., 2023b), and the diffusion-based DiffMIC-v2 (Yang et al., 2025).

On the **HAM10000** dataset, our method achieves the best performance across both metrics, surpassing the strongest baseline DiffMIC-v2 by 1.1% in Accuracy (0.894 vs. 0.883)

and 0.3% in F1-Score. This indicates that our Simplex-Aligned Diffusion strategy and Cross-Granularity Interaction module effectively capture subtle diagnostic features in skin lesions.

On the **APTOS2019** dataset, our method secures the highest Accuracy (0.848), outperforming the Transformer-based UniFormer and DiffMIC-v2. Regarding F1-score, we observe a marginal drop compared to DiffMIC-v2 (0.666 vs. 0.669). This slight dip aligns with the well-known robustness-accuracy trade-off in deep learning. The baseline likely exploits brittle, high-frequency shortcuts to maximize clean performance. By enforcing geometric consistency on the logit manifold, our Simplex-Aligned strategy acts as a strong regularizer: it suppresses reliance on these unstable features, resulting in a negligible cost on clean data but substantial gains in noise resilience. As demonstrated in Section 4.2, while other methods maintain high accuracy, they exhibit catastrophic failure under noise, whereas our method preserves robust performance.

Table 1: Quantitative comparison with state-of-the-art methods on HAM10000 and APTOS2019 datasets. The best results are highlighted in **bold**, and the second best are underlined.

| Methods | | LDAM | OHEM | MTL | DANIL | CL | ProCo | DGCMM | UniFormer | DiffMIC-v2 | Ours |
|---|---|---|---|---|---|---|---|---|---|---|---|
| **HAM10000** | Accuracy | 0.857 | 0.818 | 0.811 | 0.825 | 0.865 | 0.887 | 0.886 | 0.889 | 0.883 | **0.894** |
| | F1-Score | 0.734 | 0.660 | 0.667 | 0.674 | 0.739 | 0.763 | 0.794 | 0.802 | 0.823 | **0.826** |
| **APTOS2019** | Accuracy | 0.813 | 0.813 | 0.813 | 0.825 | 0.825 | 0.837 | 0.845 | 0.847 | 0.839 | **0.848** |
| | F1-Score | 0.620 | 0.631 | 0.632 | 0.660 | 0.652 | 0.674 | 0.685 | 0.690 | 0.669 | 0.666 |

## 4.2. Robustness Analysis

To comprehensively assess model reliability under distribution shifts, we adopted a two-fold evaluation strategy: a **Continuous Stress Test** using Gaussian noise to measure degradation dynamics, and a **Clinical Artifact Benchmarking** using specific corruptions to simulate acquisition failures in clinical settings.

In these experiments, we benchmark primarily against DiffMIC-v2, as it represents the current state-of-the-art in generative medical image classification. Our objective is to isolate the geometric stability of the generative label diffusion process itself. Therefore, DiffMIC-v2 serves as the most direct control to validate whether our Simplex-Aligned strategy effectively resolves the manifold mismatch problem inherent in prior generative classifiers.

### 4.2.1. CONTINUOUS STRESS TEST: DEGRADATION DYNAMICS

We first conduct a continuous stress test by injecting additive Gaussian noise with intensity $\sigma \in [0.05, 0.30]$ to simulate stochastic sensor degradation. Figure 2 visualizes the degradation trends on APTOS2019 and HAM10000, respectively.

**Resistance to Model Collapse.** As shown in Figure 2 (Top), on the APTOS dataset, the baseline exhibits a catastrophic collapse in agreement metrics. Specifically, as $\sigma$ increases to 0.3, its Cohen's Kappa drops precipitously from 0.73 to nearly zero (0.016), indicating that the model has degenerated to random guessing or majority-class prediction. In stark contrast, our Simplex-Aligned framework demonstrates graceful degradation, maintaining a clinically meaningful Kappa of 0.42 even under severe noise. This suggests that performing

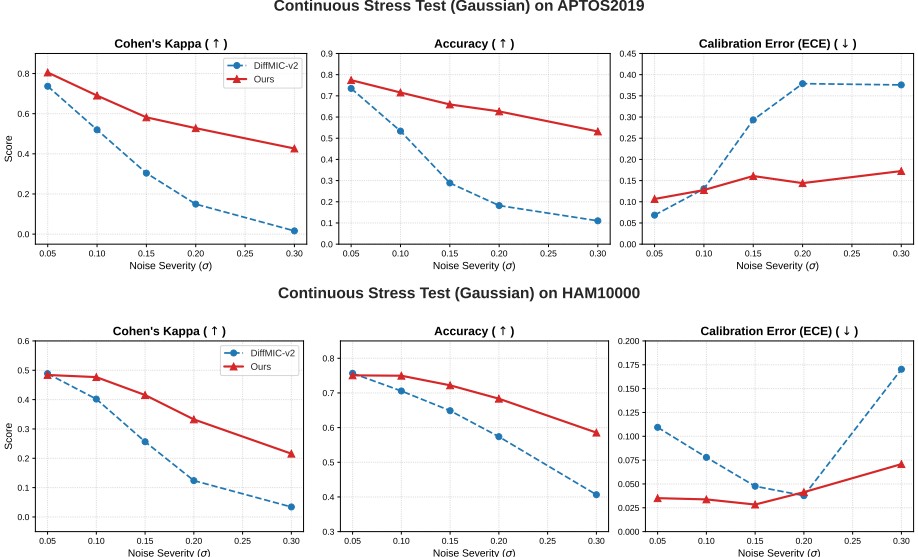

Figure 2: Continuous Stress Test Analysis. Degradation trends under Gaussian noise ($\sigma \in$ $[0.05, 0.3]$). Our method (Red) demonstrates superior stability compared to the baseline (Blue).

diffusion on the unbounded logit manifold prevents the network from being forced into incorrect simplex vertices when the input signal is ambiguous.

**Calibration Stability under Uncertainty.** A critical requirement for medical AI is reliability, knowing when the model is uncertain. Figure 2 (Bottom) highlights the calibration performance (ECE) on the HAM10000 dataset. The baseline displays a volatile U-shaped error curve, where ECE spikes significantly at high noise levels (reaching 0.17), implying that the model remains over-confident even when its predictions are wrong. Conversely, our method consistently suppresses ECE below 0.07 across the entire noise spectrum. This stability confirms that our probabilistic formulation effectively captures the epistemic uncertainty introduced by noise, ensuring that the model's confidence aligns with its actual predictive capability.

### 4.2.2. Clinical Artifact Benchmarking

Unlike general vision benchmarks that evaluate robustness across a broad spectrum of synthetic distortions (e.g., snow, fog, pixelation), medical imaging requires a strictly modality-specific evaluation protocol. Artifacts must be physically plausible within the clinical acquisition pipeline to yield meaningful robustness insights. Therefore, guided by the corruption taxonomy established by the MedMNIST-C benchmark (Di Salvo et al., 2024), we selectively utilize standard ImageNet-C (Hendrycks and Dietterich, 2019) implementations to simulate only those corruptions relevant to fundus photography and dermatoscopy.

For the **APTOS2019** dataset (Retina), we select **Shot Noise** and **Motion Blur**. We specifically employ Shot Noise (Poisson noise) to rigorously simulate the photon counting statistics inherent in low-light fundus imaging sensors. Additionally, we assess **Motion Blur**, which models the frequent artifacts caused by patient *eye saccades* or involuntary head movements during exposure, a pervasive challenge in non-mydriatic fundus photography (Di Salvo et al., 2024).

For the **HAM10000** dataset (Dermatoscopy), we prioritize motion blur and defocus blur. As dermatoscopic images are typically acquired using handheld devices, they are uniquely susceptible to artifacts caused by operator hand tremors (motion) or improper focal depth (Tschandl et al., 2018; Di Salvo et al., 2024). By benchmarking against these targeted distribution shifts, we assess the model's reliability under realistic clinical failure modes.

Table 2: Robustness Comparison on HAM10000 and APTOS2019. All metrics are reported in decimal format [0, 1]. Best results are **bolded**.

| Dataset | Noise | Sev. | Acc ↑ | | F1 ↑ | | Kappa ↑ | | ECE ↓ | |
|---|---|---|---|---|---|---|---|---|---|---|
| | | | Base | Ours | Base | Ours | Base | Ours | Base | Ours |
| HAM10000 | Defocus | 1 | 0.759 | **0.779** | 0.474 | **0.531** | 0.408 | **0.496** | 0.080 | **0.060** |
| | | 3 | 0.685 | **0.703** | 0.198 | **0.292** | 0.116 | **0.293** | 0.143 | **0.100** |
| | | 5 | 0.679 | **0.685** | 0.129 | **0.211** | 0.009 | **0.237** | 0.179 | **0.130** |
| | Motion | 1 | **0.842** | 0.826 | **0.700** | 0.657 | **0.641** | 0.636 | 0.099 | **0.073** |
| | | 3 | 0.728 | **0.735** | 0.375 | **0.442** | 0.309 | **0.384** | 0.099 | **0.058** |
| | | 5 | 0.692 | **0.696** | 0.235 | **0.281** | 0.187 | **0.279** | 0.118 | **0.115** |
| APTOS2019 | Shot | 1 | 0.140 | **0.507** | 0.113 | **0.247** | 0.112 | **0.209** | 0.405 | **0.256** |
| | | 3 | **0.499** | 0.299 | 0.151 | **0.212** | -0.005 | **0.163** | **0.176** | 0.295 |
| | | 5 | 0.324 | **0.608** | 0.176 | **0.281** | 0.062 | **0.460** | 0.224 | **0.108** |
| | Motion | 1 | 0.636 | **0.811** | 0.505 | **0.617** | 0.628 | **0.860** | 0.142 | **0.070** |
| | | 3 | 0.567 | **0.634** | 0.332 | **0.425** | 0.607 | **0.623** | 0.211 | **0.128** |
| | | 5 | 0.486 | **0.514** | 0.236 | **0.311** | **0.458** | 0.366 | 0.265 | **0.225** |

**Resilience to Acquisition Blur (HAM10000).** As shown in Table 2 (Top), our method demonstrates superior resilience against artifacts common in handheld dermatoscopy. Under **Defocus Blur**, while the baseline performance degrades rapidly at Level 5, our Simplex-Aligned model retains a Kappa of 0.237, preserving diagnostic utility even under severe out-of-focus conditions. Similarly, for **Motion Blur**, our method consistently outperforms the baseline at higher severities (Level 3-5). Notably, even in scenarios where DiffMIC-v2 achieves comparable accuracy (e.g., Motion Blur Level 1), our method yields significantly lower ECE (0.07 vs. 0.10). This indicates that our probabilistic scaling ensures the model remains well-calibrated, avoiding the over-confident but wrong predictions typical of standard diffusion models.

**Analysis of Sensor Noise and Failure Modes (APTOS2019).** The results on retinal images (Table 2, Bottom) reveal critical failure modes in the baseline that are masked by simple metrics. **Under Shot Noise**, we observe a two-stage failure in the baseline. At Level 1, the baseline's accuracy drops to 14.0%, significantly underperforming random guessing (20%). This implies hypersensitivity to noise artifacts, where the model likely

hallucinates high-frequency noise as pathological lesions. At Level 3, a striking anomaly occurs: the baseline achieves a deceptively high Accuracy of 49.9% but a *negative* Kappa (-0.01). This indicates **mode collapse**, where the network defaults to predicting only the majority class to minimize loss, effectively losing all discriminative power. In contrast, our method maintains consistent accuracy (50.7% at Level 1) and a positive Kappa, proving that it preserves structural discrimination capabilities rather than exploiting class priors. **Under Motion Blur**, simulating eye saccades, our method achieves a remarkable performance gain. At Severity 1, we outperform the baseline by over 17% in Accuracy (0.811 vs. 0.636). While the baseline struggles to resolve retinal features blurred by motion, our Simplex-Aligned diffusion robustly recovers structural semantics, demonstrating exceptional resilience to acquisition instability.

### 4.3. Ablation Study

As shown in Table 3, we investigated the contribution of each component by incrementally adding the Cross-Granularity Interaction module and the Simplex-Aligned Diffusion strategy to the baseline.

**Impact of Simplex-Alignment (Row C vs. A):** The introduction of Simplex-Alignment significantly bolsters model robustness. On APTOS2019, the accuracy under Gaussian noise jumps from 0.534 to 0.662. This confirms that constraining the diffusion process within a continuous logit simplex effectively prevents model collapse when input features are corrupted.

**Impact of Interaction Module (Row B vs. A):** The Interaction module enhances feature extraction capability, leading to improved clean accuracy on HAM10000 (0.883 to 0.891). However, relying solely on interaction (Row B) can lead to instability under noise (Noise Acc drops to 0.632 on HAM10000), suggesting that refined features require structural regularization to remain robust.

**Synergy of the Framework (Row D):** Our full model (Row D) achieves the best performance among all combination. By coupling the refined features from the Interaction module with the geometric constraints of Simplex-Alignment, we achieve a superior trade-off. Notably, on the challenging APTOS noise task, our method improves accuracy by 18.2% compared to the baseline (0.534 to 0.716), demonstrating that our components are mutually beneficial rather than redundant.

Table 3: Ablation study evaluating the contributions of Cross-Granularity Interaction and Simplex-Aligned Diffusion to classification performance and noise robustness ($\sigma = 0.1$).

| Model | Components | | HAM10000 (Derma) | | | APTOS2019 (Retina) | | |
|---|---|---|---|---|---|---|---|---|
| | Simplex | Interaction | Clean Acc | Clean F1 | Noise Acc ($\sigma = 0.1$) | Clean Acc | Clean F1 | Noise Acc ($\sigma = 0.1$) |
| A (Baseline) | × | × | 0.883 | 0.823 | 0.706 | 0.839 | **0.669** | 0.534 |
| B | × | ✓ | 0.891 | 0.821 | 0.632 | 0.834 | 0.664 | 0.714 |
| C | ✓ | × | 0.893 | 0.811 | 0.730 | 0.831 | 0.662 | 0.662 |
| **D (Ours)** | ✓ | ✓ | **0.894** | **0.826** | **0.750** | **0.848** | 0.666 | **0.716** |

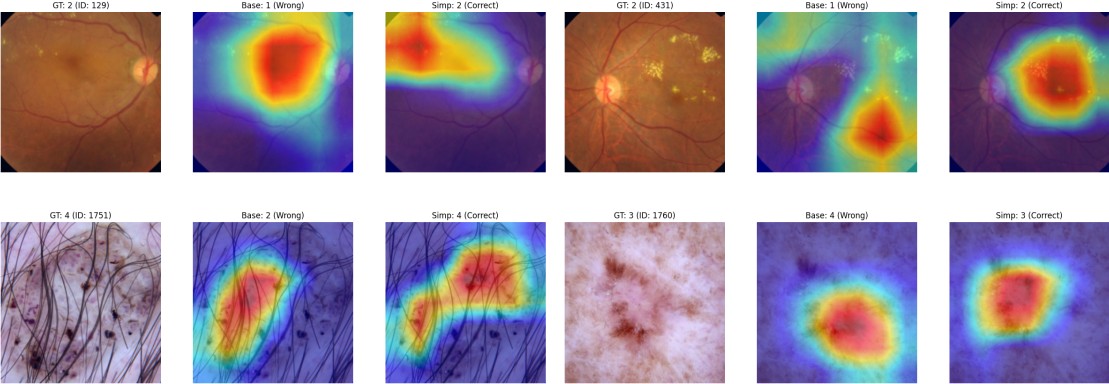

Figure 3: Qualitative Grad-CAM comparison between the baseline and our method, illustrating superior lesion localization in APTOS2019 (top) and resilience to clinical artifacts in HAM10000 (bottom).

## 4.4. Qualitative Analysis and Visual Explanations

To provide intuitive insights into the decision-making process of our proposed framework, we visualize the Grad-CAM (Selvaraju et al., 2017) attention maps in Figure 3. These visualizations compare the focus regions of the Baseline and our Simplex-Aligned method.

**Lesion Localization and Boundary Delineation (APTOS2019).** As shown in the top row of Figure 3, the retinal images present challenging pathological features, such as distinct bright lesions known as **Hard Exudates** and **Cotton Wool Spots**. The baseline model tends to generate diffused attention maps, often confusing optical artifacts (e.g., light reflections or the optic disc) with actual lesions. In contrast, our Simplex-Aligned method demonstrates superior semantic selectivity. It accurately distinguishes pathological boundaries, focusing precisely on the clusters of Hard Exudates while suppressing background noise. This precise localization indicates that our method learns robust features on the logit manifold rather than overfitting to global image statistics.

**Robustness to Occlusion and Shape Consistency (HAM10000).** The bottom row illustrates the model's performance on dermatoscopic images, which are frequently compromised by artifacts such as hair occlusion and ruler markings. For **Class 4 (Melanocytic nevi)**, where the lesion is heavily occluded by dense hair, the baseline's attention is disrupted, tracking the hair strands instead of the pigment network. Our method, however, exhibits remarkable robustness to such occlusions, successfully bypassing the hair artifacts to focus on the underlying lesion patterns. Similarly, for **Class 3 (Benign keratosis-like lesions)**, which typically present with irregular borders, our method captures the *entire* extent of the lesion, delineating a larger and more accurate boundary compared to the baseline's center-biased focus. This confirms that our method can effectively learn the full morphological structure of the disease across different classes. We provide more visualization results covering additional classes and failure cases in Appendix D.

### 4.5. Computational Efficiency and Practical Overhead

Diffusion-based classifiers inherently involve iterative inference, which introduces additional computational overhead compared to standard discriminative CNNs. However, our Simplex-Aligned Diffusion significantly reduces this overhead relative to prior diffusion-based medical classifiers by operating in a low-dimensional logit space rather than the high-dimensional pixel space.

As shown in Table 4, our method achieves an inference latency of 2.40 ms per image, which is slightly lower than DiffMIC-v2 (2.51 ms), while reducing the total computational cost by approximately 39% in GFLOPs (168.8G vs. 278.0G) and requiring less peak GPU memory.

Although standard CNNs remain faster in absolute terms, our goal is not to match their raw throughput, but to substantially reduce the overhead of diffusion-based classifiers while achieving improved robustness and calibration under acquisition shifts. In this context, the additional $\sim$2 ms latency represents a favorable trade-off for safety-critical medical applications, where reliability and calibrated uncertainty are essential.

Table 4: Computational efficiency and hardware overhead comparison.

| Metric | ResNet-50 | DiffMIC-v2 | Ours |
|---|---|---|---|
| Inference Latency (ms/img) | 0.36 | 2.51 | 2.40 |
| Total GFLOPs | 2.05G | 277.99G | 168.80G |
| Peak GPU Memory (GB) | $\sim$0.90 | 2.37 | 1.95 |
| Throughput (FPS) | $\sim$2700 | $\sim$398 | $\sim$416 |

## 5. Conclusion

We presented Simplex-Aligned Diffusion, a generative classifier that resolves the geometric conflict between Gaussian noise and discrete simplices by operating on the unbounded logit manifold. Integrated with a Cross-Granularity Interaction module, our framework achieves superior noise robustness and calibration on APTOS2019 and HAM10000 while maintaining competitive accuracy.

Crucially, the improved calibration directly enhances clinical decision reliability, as the model's confidence scores serve as a trustworthy proxy for diagnostic accuracy, thereby reducing the risk of overconfident misdiagnosis. Our qualitative analysis further reveals that the model maintains a "meaningful failure" mode under extreme noise; even in cases of incorrect classification, the framework preserves persistent visual localization of pathological regions, whereas baselines often lose semantic focus.

While our current validation focuses on 2D benchmarks, the Simplex-Aligned formulation is inherently modality-agnostic. Future work will explore its application to 3D medical volumes (e.g., CT/MRI) (Xie et al., 2026) and 1D signals, alongside acceleration techniques like consistency distillation and flow matching to facilitate real-time clinical deployment.

## Acknowledgments

The work was supported by the US NSF CAREER award IIS-2239537.

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

## Appendix A. Implementation Details and Reproducibility

### A.1. Implementation Details.

The framework is implemented in PyTorch. For the Transformer-Enhanced Visual Tokenizer, we utilize ResNet-18/50 as the backbone for both global and local streams, initialized with ImageNet pre-trained weights. The interaction module employs a standard Transformer encoder layer with a hidden dimension of 512. For the Simplex-Aligned Diffusion, we set the total diffusion timesteps $T = 1000$ with a linear noise schedule from $\beta_1 = 10^{-4}$ to $\beta_T = 0.02$. The simplex scaling factor $\lambda$ is empirically set to $1.5 \log C$. The model is trained using the AdamW optimizer with an initial learning rate of $5 \times 10^{-4}$, decayed via a cosine annealing schedule. We train for 1,000 epochs with a batch size of 64. All experiments are conducted on 3 NVIDIA H100 GPUs.

## A.2. Dataset Preprocessing and Splitting

To ensure a fair and rigorous comparison, we align our data partitioning protocols with the DiffMIC-v2 benchmark (Yang et al., 2025).

- **HAM10000:** We adhere to the standard 7:3 split ratio for training and testing, consistent with the protocol in (Gong et al., 2020; Yang et al., 2025). This yields 7,010 images for training/validation and 3,005 images for testing.

- **APTOS2019** Following the baseline settings (Yang et al., 2025), we employ a 7:3 split on the official dataset (2,929 for training/validation, 733 for testing).

**Rigorous Model Selection Strategy:** It is important to note that the original DiffMIC-v2 implementation performs model selection by monitoring performance directly on the test set (i.e., reporting the best-epoch results on the test split). To avoid such test-set leakage and ensure clinical validity, we implement a stricter evaluation protocol: we randomly hold out 10% of the training set as an independent validation set for hyperparameter tuning and checkpoint selection. All reported metrics in our main paper are derived from the unseen test set using the fixed model selected via the validation set.

All images undergo a standardized pre-processing pipeline:

- **Resize & Crop:** Images are center-cropped and resized to $224 \times 224$ to maintain consistent input resolution.

- **Normalization:** We apply Z-score normalization using standard ImageNet statistics: mean $\mu = [0.485, 0.456, 0.406]$ and std $\sigma = [0.229, 0.224, 0.225]$.

- **Augmentation:** During training, we apply random horizontal flips and mild rotations ($\pm 10°$) to mitigate overfitting. No augmentation is applied during inference.

## A.3. Architecture and Backbone Selection

A critical deviation in our implementation compared to the original DiffMIC-v2 (Yang et al., 2025) is the choice of the visual backbone. While the original DiffMIC-v2 employs EfficientSAM (Xiong et al., 2024) as the image encoder, we standardize our backbone to ResNet-50 for both the baseline and our method. This decision is driven by two key factors:

1. **Resolution Mismatch:** EfficientSAM is natively designed for high-resolution inputs ($1024 \times 1024$). However, standard medical image classification benchmarks (e.g., ISIC, APTOS) are typically evaluated at $224 \times 224$. Forcing a $224 \times 224$ input into a $1024 \times 1024$ model requires aggressive interpolation, which introduces artificial sub-pixel artifacts and does not reflect real-world clinical deployment constraints.

2. **Fairness in Comparison:** Our objective is to validate the effectiveness of the Simplex-Aligned Diffusion strategy, rather than the power of the feature extractor. By using a standard ResNet-50, we ensure that performance gains are attributable solely to our methodological contributions (Logit-space diffusion and Cross-Granularity Interaction) rather than a larger backbone capacity.

## A.4. Training Configuration

All models are implemented in PyTorch and trained on NVIDIA H100 GPUs. We use the AdamW optimizer with a cosine annealing learning rate schedule. Detailed hyperparameters are listed in Table 5.

Table 5: Hyperparameter settings for training.

| Parameter | Value |
|---|---|
| Image Size | $224 \times 224$ |
| Batch Size | 64 |
| Learning Rate | $1 \times 10^{-4}$ |
| Weight Decay | $1 \times 10^{-4}$ |
| Diffusion Timesteps $(T)$ | 1000 |
| Noise Schedule | Linear $(\beta_1 = 10^{-4}, \beta_T = 0.02)$ |
| Total Epochs | 1000 |

## A.5. Training and Inference Algorithm

---
**Algorithm 1:** Training of Simplex-Aligned Diffusion

---
**Input:** Training images $\mathcal{D} = \{(\mathbf{x}^{(i)}, \mathbf{y}^{(i)})\}_{i=1}^{N}$, Total timesteps $T$
**Output:** Optimized model parameters $\theta$

$\mathbf{z}_0 \leftarrow \text{CenterLogRatio}(\mathbf{y})$ // Map one-hot label to Logit Space

**for** *each iteration* **do**
    Sample batch $(\mathbf{x}, \mathbf{z}_0)$ from $\mathcal{D}$
    Sample timestep $t \sim \text{Uniform}(\{1, \dots, T\})$
    Sample noise $\boldsymbol{\epsilon} \sim \mathcal{N}(\mathbf{0}, \mathbf{I})$
    $\mathbf{z}_t \leftarrow \sqrt{\bar{\alpha}_t}\mathbf{z}_0 + \sqrt{1 - \bar{\alpha}_t}\boldsymbol{\epsilon}$ // Add noise
    $\mathbf{v}_g, \mathbf{v}_l, \mathbf{Z}_{\text{raw}} \leftarrow \text{DualStreamEncoder}(\mathbf{x})$ // Extract Priors & Tokens
    $\mathbf{F}_{\text{ref}} \leftarrow \text{TransformerInteraction}(\mathbf{Z}_{\text{raw}})$ // Semantic Refinement
    $\mathcal{M} \leftarrow \text{ConstructMap}(\mathbf{v}_g, \mathbf{v}_l)$ // Spatial Guidance Construction
    $\boldsymbol{\epsilon}_\theta \leftarrow \text{UNet}(\mathbf{z}_t, t, \mathcal{M}, \mathbf{F}_{\text{ref}})$ // Predict noise
    $\mathcal{L}_{\boldsymbol{\epsilon}} \leftarrow \|\boldsymbol{\epsilon} - \boldsymbol{\epsilon}_\theta\|^2$
    Update $\theta$ using $\nabla_\theta \mathcal{L}_{\boldsymbol{\epsilon}}$
**end**

---

---

**Algorithm 2:** Inference / Sampling Procedure

---

**Input:** Test Image $\mathbf{x}$, Sampling timesteps $T$
**Output:** Predicted Class Probability $\hat{\mathbf{p}}$

$\mathbf{z}_T \sim \mathcal{N}(\mathbf{0}, \mathbf{I})$                      // Initialize from standard Gaussian
$\mathbf{v}_g, \mathbf{v}_l, \mathbf{Z}_{\text{raw}} \leftarrow \text{DualStreamEncoder}(\mathbf{x})$        // Extract Priors & Raw Tokens
$\mathbf{F}_{\text{ref}} \leftarrow \text{TransformerInteraction}(\mathbf{Z}_{\text{raw}})$           // Semantic Refinement
$\mathcal{M} \leftarrow \text{ConstructMap}(\mathbf{v}_g, \mathbf{v}_l)$               // Spatial Guidance

**for** $t = T, \ldots, 1$ **do**
    $\boldsymbol{\epsilon}_{\text{pred}} \leftarrow \text{UNet}(\mathbf{z}_t, t, \mathcal{M}, \mathbf{F}_{\text{ref}})$
    $\mathbf{z}_{t-1} \leftarrow \frac{1}{\sqrt{\alpha_t}}\left(\mathbf{z}_t - \frac{1-\alpha_t}{\sqrt{1-\bar{\alpha}_t}}\boldsymbol{\epsilon}_{\text{pred}}\right) + \sigma_t \mathbf{z}_{\text{noise}}$
**end**

$\hat{\mathbf{p}} \leftarrow \text{Softmax}(\lambda \cdot \mathbf{z}_0)$             // Project Logits to Probability Simplex
**return** $\hat{\mathbf{p}}$

---

## Appendix B. Training Dynamics and Convergence Analysis

To demonstrate the stability of our proposed framework, we compare the validation metric curves of the Baseline (DiffMIC-v2) and our method over 1000 epochs.

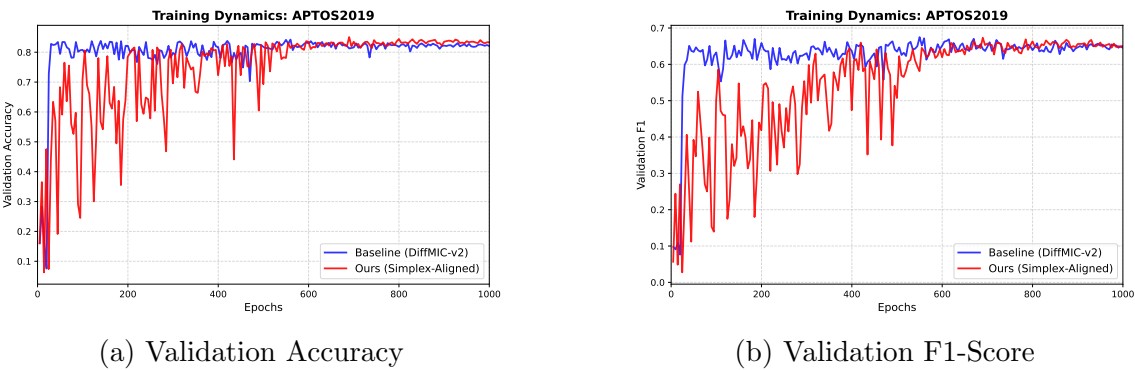

(a) Validation Accuracy                      (b) Validation F1-Score

Figure 4: Comparison of Training Dynamics. The validation curves for (a) Accuracy and (b) F1-Score. The baseline (Blue) converges rapidly in the early epochs but suffers from saturation. In contrast, our method (Red) demonstrates continuous improvement.

As illustrated in Figure 4, the baseline model exhibits a phenomenon of **Early Saturation**. Since it forces the diffusion model to approximate discrete one-hot vectors directly, the model quickly memorizes easy samples but fails to learn robust features for hard samples, leading to metric fluctuations or a decline in later stages. Conversely, our Simplex-Aligned model operates in a continuous logit space, enabling finer-grained optimization. Even in the mid-to-late stages of training, our curve continues to rise, indicating that although optimization in the logits space progresses more slowly than in the one-hot space, the model

does not prematurely saturate or fall into ambiguous local optima. Instead, it continues to explore more informative regions of the parameter space. This sustained exploration not only leads to competitive final performance but also enhances robustness to noise. In contrast, the baseline model quickly enters a saturated regime, where later training updates become ineffective. These empirical observations align perfectly with our theoretical analysis in Section F. Specifically, the baseline's volatility corroborates the existence of the systematic bias $\delta$ (Proposition 1), which causes target jittering during training, whereas our method's training dynamics validates the target consistency of the unconstrained logit diffusion.

## Appendix C. Extended Robustness Evaluation

### C.1. Visualization of Clinical Corruptions

To provide an intuitive context for the domain-specific robustness evaluation, we visualize the simulated artifacts on both datasets. Figures 5 and 6 display the degradation effects across increasing severity levels (Level 1, 3, and 5) for APTOS 2019 and HAM10000, respectively.

### C.2. Full Benchmarking Tables

Due to space constraints in the main text, we present the complete benchmarking results for all evaluated corruption types and severity levels.

Table 6: Comparison of other noise types on the APTOS2019 dataset.

| Type | Sev | Acc ↑ | | F1 ↑ | | Kappa ↑ | | ECE ↓ | |
|------|-----|-------|------|------|------|---------|------|-------|------|
| | | Ours | Base | Ours | Base | Ours | Base | Ours | Base |
| | 1 | 0.561 | 0.625 | 0.383 | 0.450 | 0.593 | 0.495 | 0.206 | 0.135 |
| Defocus | 3 | 0.219 | 0.176 | 0.118 | 0.156 | 0.105 | 0.026 | 0.634 | 0.612 |
| | 5 | 0.197 | 0.138 | 0.106 | 0.119 | 0.060 | 0.002 | 0.661 | 0.651 |
| | 1 | 0.755 | 0.716 | 0.481 | 0.414 | 0.772 | 0.725 | 0.109 | 0.163 |
| saturate | 3 | 0.788 | 0.753 | 0.615 | 0.570 | 0.875 | 0.857 | 0.068 | 0.098 |
| | 5 | 0.771 | 0.731 | 0.604 | 0.556 | 0.802 | 0.805 | 0.074 | 0.113 |
| | 1 | 0.826 | 0.787 | 0.626 | 0.606 | 0.883 | 0.835 | 0.045 | 0.069 |
| brightness | 3 | 0.679 | 0.428 | 0.503 | 0.349 | 0.674 | 0.383 | 0.079 | 0.348 |
| | 5 | 0.543 | 0.294 | 0.375 | 0.215 | 0.516 | 0.166 | 0.188 | 0.489 |
| | 1 | 0.794 | 0.769 | 0.537 | 0.439 | 0.844 | 0.762 | 0.086 | 0.120 |
| contrast | 3 | 0.708 | 0.699 | 0.405 | 0.380 | 0.701 | 0.579 | 0.102 | 0.128 |
| | 5 | 0.491 | 0.461 | 0.266 | 0.280 | 0.341 | 0.286 | 0.147 | 0.243 |

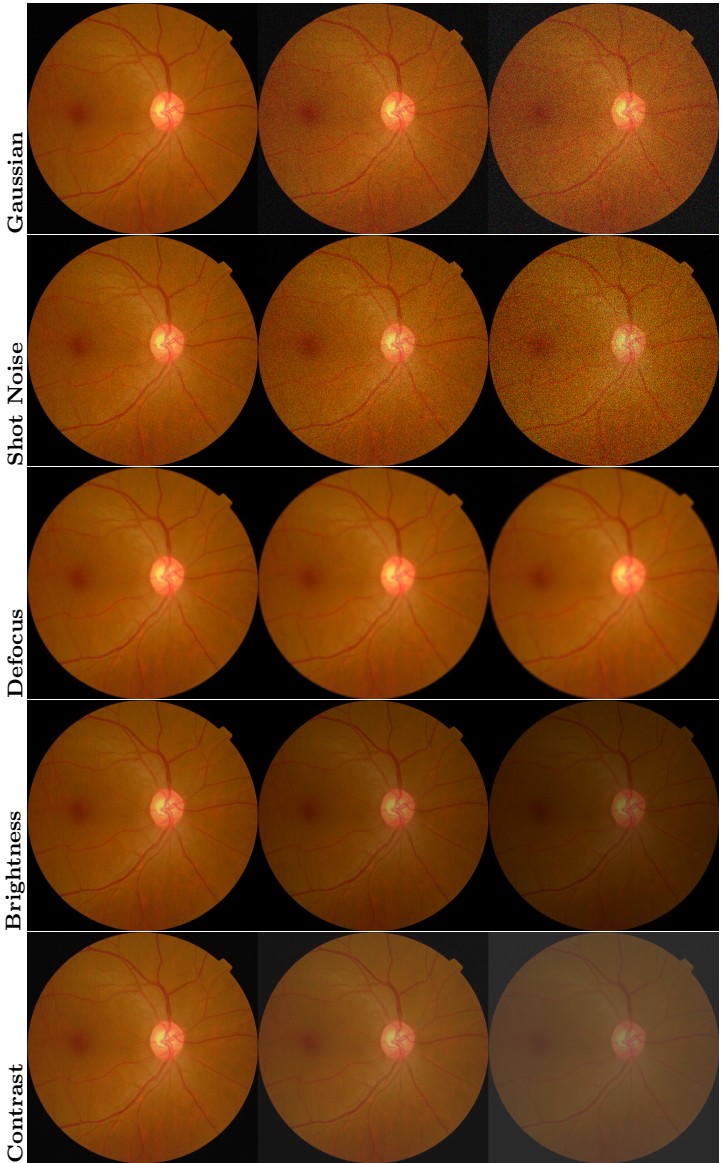

Figure 5: Clinical Corruptions on APTOS2019 (Retina). **Shot Noise** simulates photon starvation; **Defocus Blur** mimics lens errors; **Brightness/Contrast** represent exposure instabilities.

## Appendix D.  Additional Qualitative Results

We provide extensive qualitative comparisons to further substantiate the robustness of our Simplex-Aligned Diffusion framework.

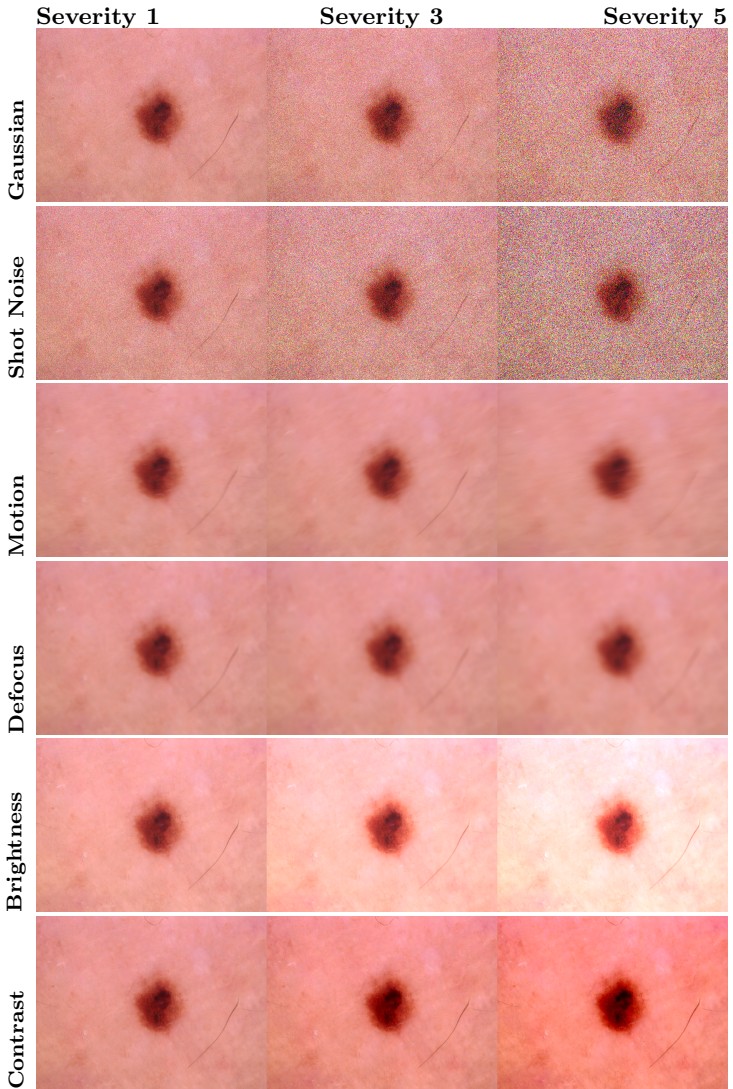

Figure 6: Clinical Corruptions on HAM10000 (Dermatoscopy).

## Appendix E. Mathematical Definitions

### E.1. Cohen's Kappa ($\kappa$)

Cohen's Kappa measures the agreement between the predicted classification and the ground truth, correcting for chance agreement. It is defined as:

$$\kappa = \frac{p_o - p_e}{1 - p_e} \tag{6}$$

where $p_o$ is the relative observed agreement (Accuracy), and $p_e$ is the hypothetical probability of chance agreement. For ordinal tasks like APTOS, we utilize the Quadratic Weighted Kappa.

Table 7: Comparison of other noise types on the HAM10000 dataset.

| Type | Sev | Acc ↑ | | F1 ↑ | | Kappa ↑ | | ECE ↓ | |
| | | Ours | Base | Ours | Base | Ours | Base | Ours | Base |
|---|---|---|---|---|---|---|---|---|---|
| saturate | 1 | 0.683 | 0.666 | 0.304 | 0.387 | 0.221 | 0.294 | 0.040 | 0.086 |
| | 3 | 0.692 | 0.668 | 0.427 | 0.333 | 0.284 | 0.209 | 0.051 | 0.067 |
| | 5 | 0.539 | 0.317 | 0.151 | 0.110 | 0.094 | 0.0172 | 0.117 | 0.242 |
| shot_noise | 1 | 0.711 | 0.122 | 0.301 | 0.036 | 0.363 | 0.001 | 0.076 | 0.569 |
| | 3 | 0.103 | 0.111 | 0.044 | 0.029 | 0.033 | 0.001 | 0.604 | 0.454 |
| | 5 | 0.051 | 0.096 | 0.014 | 0.062 | 0.001 | 0.179 | 0.718 | 0.188 |
| brightness | 1 | 0.867 | 0.855 | 0.752 | 0.721 | 0.750 | 0.732 | 0.135 | 0.090 |
| | 3 | 0.754 | 0.737 | 0.493 | 0.423 | 0.516 | 0.437 | 0.062 | 0.065 |
| | 5 | 0.720 | 0.698 | 0.390 | 0.270 | 0.372 | 0.234 | 0.035 | 0.114 |
| contrast | 1 | 0.742 | 0.713 | 0.443 | 0.381 | 0.352 | 0.381 | 0.081 | 0.106 |
| | 3 | 0.681 | 0.663 | 0.143 | 0.163 | 0.047 | 0.130 | 0.089 | 0.120 |
| | 5 | 0.678 | 0.580 | 0.1156 | 0.114 | 0.002 | 0.000 | 0.175 | 0.177 |

### E.2. Expected Calibration Error (ECE)

ECE measures the expected discrepancy between the model's confidence and its empirical accuracy. We partition the $n$ test samples into $M$ equally spaced bins (e.g., $M = 15$) based on their prediction confidence. For a sample $i$, let $\hat{y}_i$ be the predicted class and $\hat{p}_i = \max_c P(y = c|x_i)$ be the associated confidence. Let $B_m$ be the set of samples falling into the $m$-th bin. ECE is calculated as the weighted average of the absolute difference between accuracy and confidence:

$$\text{ECE} = \sum_{m=1}^{M} \frac{|B_m|}{n} \left| \text{acc}(B_m) - \text{conf}(B_m) \right| \tag{7}$$

where $|B_m|$ is the number of samples in bin $m$, $\text{acc}(B_m) = \frac{1}{|B_m|} \sum_{i \in B_m} \mathbb{1}(\hat{y}_i = y_i)$ is the bin's accuracy ($y_i$ is the ground truth), and $\text{conf}(B_m) = \frac{1}{|B_m|} \sum_{i \in B_m} \hat{p}_i$ is the average confidence. Lower ECE indicates a better-calibrated model, which is vital for reliable clinical decision-making.

## Appendix F. Theoretical Analysis

In this section, we analyze the theoretical limitations of applying standard Gaussian diffusion directly to simplex-constrained data (e.g., One-Hot labels). We verify two hypotheses corresponding to common baselines: first, that unconstrained Gaussian diffusion leads to a systematic boundary bias due to support mismatch; second, that strictly enforcing constraints step-wise renders the training objective intractable.

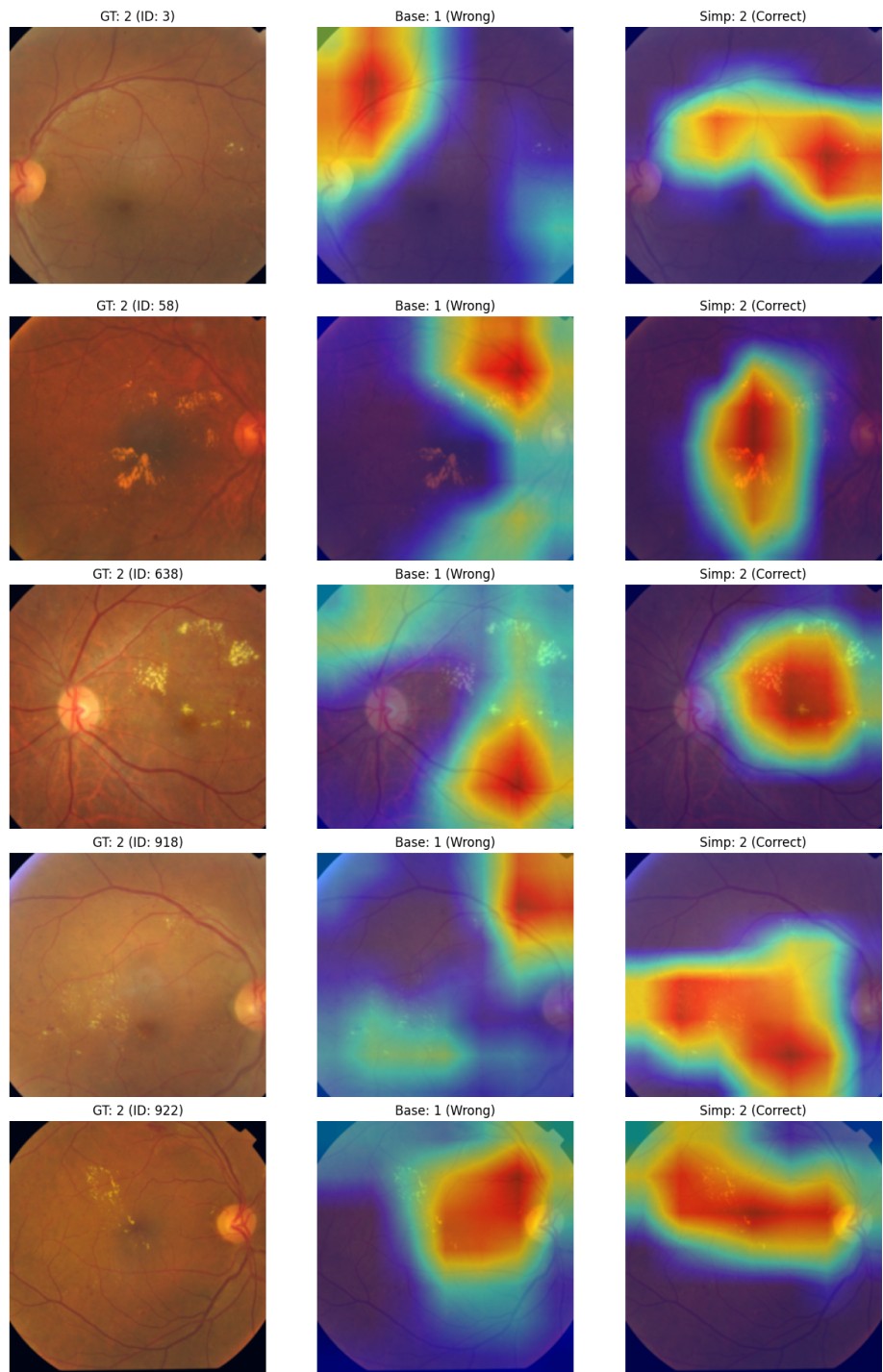

Figure 7: Extended Qualitative Analysis on APTOS2019 (Retina).

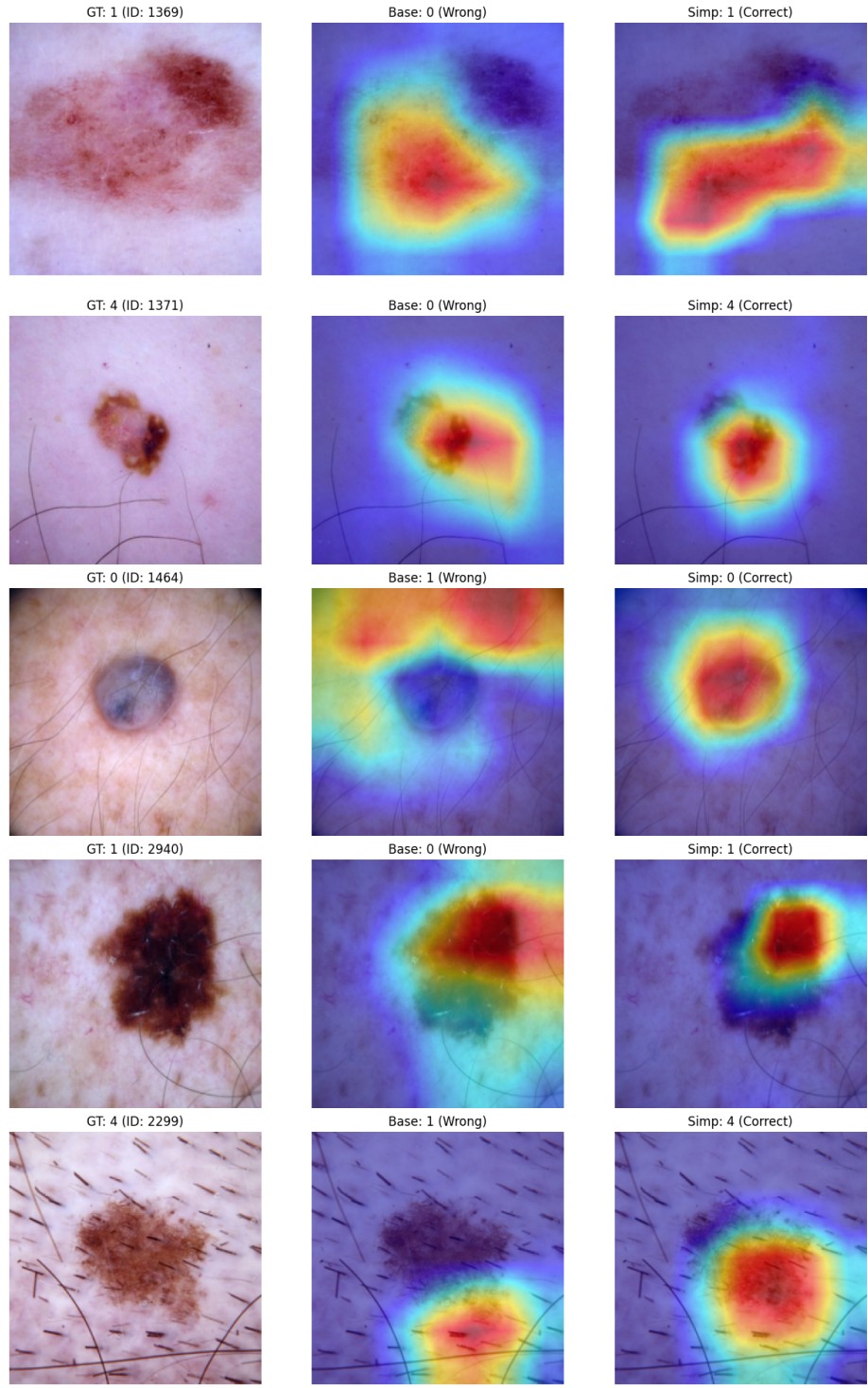

Figure 8: Extended Qualitative Analysis on HAM10000 (Dermatoscopy).

**F.1. Bias from Probability Leakage and Rectification**

**Proposition 1** *Standard DDPM defines the reverse process posterior as an unconstrained Gaussian distribution supported on $\mathbb{R}^C$. However, valid One-Hot data lies strictly on the simplex boundary. We show that the Gaussian assumption inherently allocates probability mass to invalid regions (Probability Leakage), leading to a systematic bias when the unconstrained mean is used to estimate valid data.*

**Proof Step 1: The Unbounded Gaussian Posterior.** By definition of the forward diffusion process, the true posterior $q(\mathbf{x}_{t-1}|\mathbf{x}_t, \mathbf{x}_0)$ is derived as a standard Gaussian over the Euclidean space:

$$q(\mathbf{x}_{t-1}|\mathbf{x}_t, \mathbf{x}_0) = \mathcal{N}(\mathbf{x}_{t-1}; \tilde{\boldsymbol{\mu}}_t, \tilde{\beta}_t \mathbf{I}) \tag{8}$$

The standard MSE objective trains the model to approximate this unbounded mean $\tilde{\boldsymbol{\mu}}_t$.

   **Step 2: Probability Leakage.** Since the target $\mathbf{x}_0$ lies on the simplex (One-Hot), the valid signal must be non-negative ($\mathbf{x} \geq 0$). However, the Gaussian posterior spreads probability mass into the invalid negative half-space. We define this *Probability Leakage* as:

$$P_{\text{leak}} = \int_{-\infty}^{0} \mathcal{N}(x; \tilde{\mu}, \tilde{\beta}) \, dx > 0 \tag{9}$$

Because $P_{\text{leak}} > 0$, the standard Gaussian mean $\tilde{\boldsymbol{\mu}}_t$ is no longer a valid estimator for the constrained data.

   **Step 3: Derivation of Systematic Bias.** To recover a valid estimate, we must evaluate the expectation over the valid domain ($x \geq 0$). This is equivalent to calculating the first moment of a Rectified (Truncated) Gaussian, which normalizes the remaining probability mass $(1 - P_{\text{leak}})$:

$$\mathbb{E}_{\text{valid}}[x] = \frac{1}{1 - P_{\text{leak}}} \int_{0}^{\infty} x \cdot \mathcal{N}(x; \tilde{\mu}, \tilde{\beta}) \, dx \tag{10}$$

Solving this integral reveals a shift from the original mean:

$$\mathbb{E}_{\text{valid}}[x] = \tilde{\mu} + \underbrace{\sqrt{\tilde{\beta}} \cdot \lambda\left(\frac{-\tilde{\mu}}{\sqrt{\tilde{\beta}}}\right)}_{\text{Bias } \delta} \tag{11}$$

where $\lambda(\cdot) = \frac{\phi(\cdot)}{1 - \Phi(\cdot)}$ is the Inverse Mills Ratio. Note that the denominator $1 - \Phi(\cdot)$ is exactly the valid probability mass $(1 - P_{\text{leak}})$.

   **Step 4: Conclusion.** The standard DDPM minimizes error towards $\tilde{\boldsymbol{\mu}}_t$, ignoring the bias term $\boldsymbol{\delta}$ required to compensate for probability leakage. Since $\boldsymbol{\delta} > 0$, the model systematically underestimates the values needed to stay on the manifold, causing the generated samples to drift away from the simplex vertices (over-smoothing) towards the interior. ∎

### F.2. Intractability of Projected Diffusion

**Proposition 2** *An alternative baseline strategy is to strictly enforce simplex constraints via a normalization function $f(\cdot)$ (e.g., Softmax) at each forward diffusion step. We show that this approach breaks the Gaussian marginal property, causing the standard training objective to rely on a biased approximation.*

**Proof  Step 1: The Projected Forward Process.** Consider a process where a non-linear projection $f : \mathbb{R}^C \to \Delta^{C-1}$ is applied immediately after noise injection at every transition step:

$$\mathbf{x}_t = f(\sqrt{1 - \beta_t}\mathbf{x}_{t-1} + \sqrt{\beta_t}\boldsymbol{\epsilon}_t) \tag{12}$$

**Step 2: Loss of Closed-Form Marginals.** Standard DDPM efficiency relies on the Gaussian superposition property, allowing sampling of $\mathbf{x}_t$ directly from $\mathbf{x}_0$ in $O(1)$ time. However, in the projected process, the state $\mathbf{x}_t$ depends on $\mathbf{x}_{t-1}$, which is itself a non-linear function of $\mathbf{x}_{t-2}$. Unrolling this recursion yields a nested composition of non-linearities:

$$\mathbf{x}_t = f\left(\sqrt{1 - \beta_t}f(\dots) + \sqrt{\beta_t}\boldsymbol{\epsilon}_t\right) \tag{13}$$

Unlike the linear case, these nested functions do not collapse into a single Gaussian distribution. Consequently, the marginal $q(\mathbf{x}_t|\mathbf{x}_0)$ has no analytical closed form, making the true posterior intractable.

**Step 3: Systematic Bias from Approximation Gap.** To bypass this intractability, baselines typically approximate the objective by targeting the *projection of the expectation*. However, the true optimal denoising target corresponds to the *expectation of the projected states*. Due to the strict non-linearity of $f$, the expectation operation does not commute with the projection mapping (i.e., $\mathbb{E}[f] \neq f(\mathbb{E})$). This introduces a systematic discrepancy $\mathcal{J}_t$:

$$\mathcal{J}_t = \left\| \underbrace{\mathbb{E}_{\boldsymbol{\epsilon}}[f(\mathbf{u}_t + \boldsymbol{\epsilon})]}_{\text{True Denoising Target}} - \underbrace{f(\mathbb{E}_{\boldsymbol{\epsilon}}[\mathbf{u}_t + \boldsymbol{\epsilon}])}_{\text{Biased Proxy Target}} \right\| > 0 \tag{14}$$

where $\mathbf{u}_t$ denotes the pre-projection hidden state. We define this residue $\mathcal{J}_t$ as the **Approximation Gap** (analogous to Jensen's Gap). Standard diffusion training explicitly optimizes towards the Biased Proxy Target. Since $\mathcal{J}_t$ is non-zero, the model is optimizing a mis-specified objective, where the learned mean is structurally shifted away from the true data manifold. This error accumulates over timesteps, preventing convergence to the valid simplex distribution. ∎

## Appendix G.  Visualization of Failure Modes and Explainability

**Defining "Meaningful Failure".**  We define a "Meaningful Failure" as a scenario where the model's categorical prediction is incorrect, yet its internal spatial reasoning, as evidenced by Grad-CAM localization, remains semantically aligned with the actual pathological regions. This distinction is critical for clinical safety: a model that fails while still highlighting the correct lesion is far more trustworthy than one that fails by shifting its focus to background artifacts.

**Dataset-Specific Failure Analysis.** In this analysis, we specifically focus on **Motion Blur for HAM10000** and **Shot Noise for APTOS2019**:

- **HAM10000 (Motion Blur):** As handheld dermatoscopy is uniquely susceptible to operator hand tremors, Motion Blur represents the most pervasive clinical artifact in skin lesion screening.

- **APTOS2019 (Shot Noise):** As noted in our statistical evaluation (Table 11), Shot Noise Level 5 exhibits the highest performance variance. Rather than overlooking this instability, we utilize visualization to investigate its cause. We observe that extreme Poisson noise creates a high-variance regime where single-seed outliers may occur, yet our model consistently preserves structural discrimination even when the label prediction drifts.

**Logit Variance as Diagnostic Intent.** We interpret the **Variance of Logits (Var)** as a quantitative measure of the model's **diagnostic intent**. A high variance indicates that the model is actively discriminating between classes, forcing the diffusion process toward a specific vertex of the logit manifold. In contrast, the baseline's low variance (often $< 0.1$) indicates "logit flattening," where the model loses its ability to distinguish pathology from noise. This leads to **mode collapse**, where the baseline defaults to the majority class regardless of the visual input. By maintaining high discriminative variance, our Simplex-Aligned Diffusion demonstrates that it continues to "search" for lesions under noise levels that cause standard models to fail blindly. Through Grad-CAM visualization of Figure 10 and Figure 9, we observe that our Simplex-Aligned Diffusion consistently maintains precise sickness localization, even when the specific diagnostic label is misidentified. This indicates that our logit-space geometric regularizer effectively preserves the semantic integrity of the latent features. In contrast, the baseline (DiffMIC-v2) typically exhibits **mode collapse**, defaulting to the majority class with sparse or chaotic attention maps that fail to focus on diagnostic regions. This demonstrates that our method is more clinically reliable: it provides persistent visual guidance to clinicians even when the signal-to-noise ratio is severely degraded, whereas the baseline's failure is semantically uninformative.

## Appendix H. Sensitivity Analysis

To investigate the robustness of our Simplex-Aligned Diffusion framework, we conducted extensive ablation studies on two key hyperparameters: the scaling factor $\lambda$ and the label smoothing constant $\epsilon$. The results are summarized in Table 8.

**Scaling factor $\lambda$.** As shown in Table 8, model performance exhibits a clear single-peak behavior with respect to the scaling factor $\lambda$, achieving the best results at $\lambda = 1.5$ on both benchmarks. When moving away from this value, performance degrades smoothly rather than collapsing abruptly. Smaller $\lambda$ values under-utilize the simplex alignment, while excessively large values lead to more over-confident predictions, which slightly harms robustness on ambiguous samples. Overall, the observed trend indicates that the method is not overly sensitive to $\lambda$, and a fixed $\lambda = 1.5$ provides a stable and reproducible operating point.

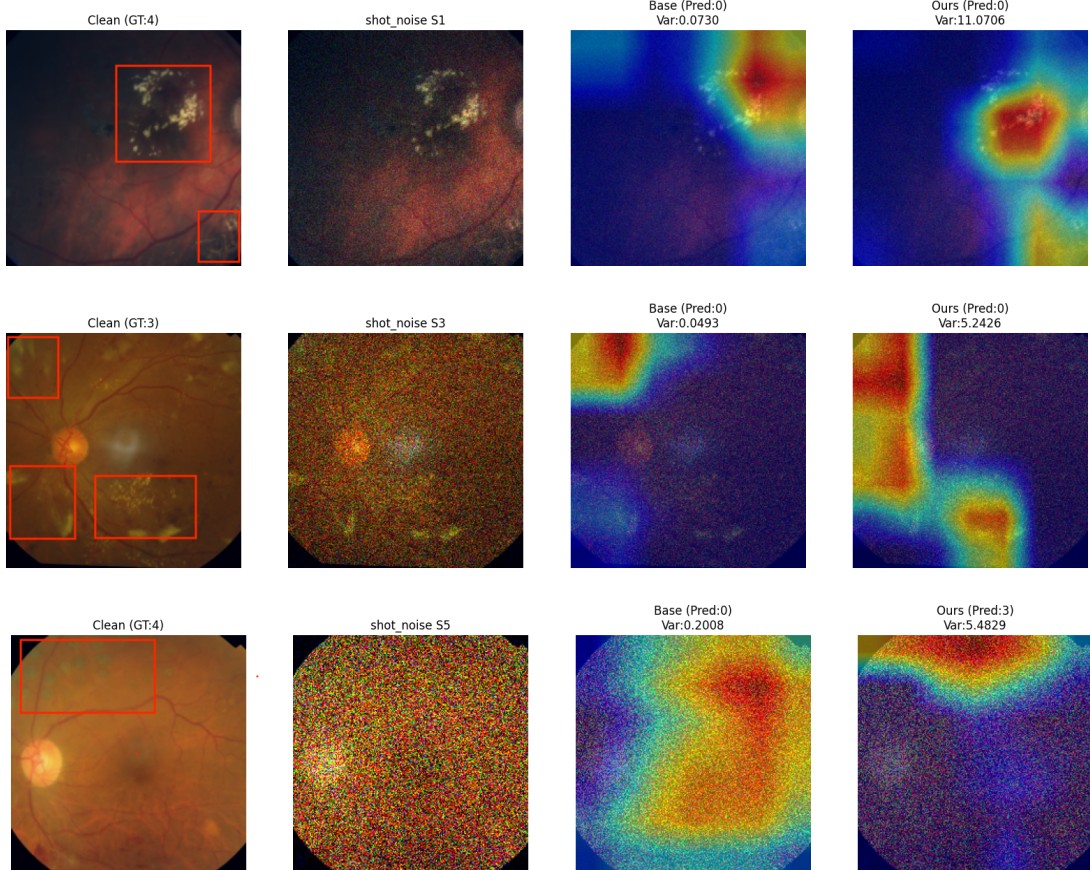

Figure 9: Qualitative comparison of failure modes for APTOS2019 under severe clinical artifacts (shot noise). **Left to Right:** Clean image (GT), Noisy image, Baseline Grad-CAM, and Our Simplex-Aligned Grad-CAM. Even when both models output incorrect labels, our method maintains a high discriminative variance (Var) and consistently localizes the sick area, whereas the baseline attention becomes chaotic or collapses to the majority class.

**Label smoothing $\epsilon$.** For label smoothing, $\epsilon = 10^{-3}$ consistently yields the best accuracy across both datasets. Larger $\epsilon$ ($10^{-2}$) overly softens the label distribution and reduces discriminative power, whereas smaller $\epsilon$ ($10^{-4}$) approaches hard one-hot labels and weakens the numerical stability of the CLR transformation. Importantly, performance varies smoothly across the tested range, suggesting that the model remains robust within reasonable smoothing strengths.

**Learnable scaling.** We considered making $\lambda$ a learnable parameter, but opted for a fixed global scaling in our final design. Since $\lambda$ directly controls the geometry of the simplex-aligned logit transformation, learning it jointly with the model can introduce additional optimization instability and reduce interpretability. Given the smooth sensitivity trends, a fixed $\lambda$ offers a favorable balance between performance, stability, and ease of deployment.

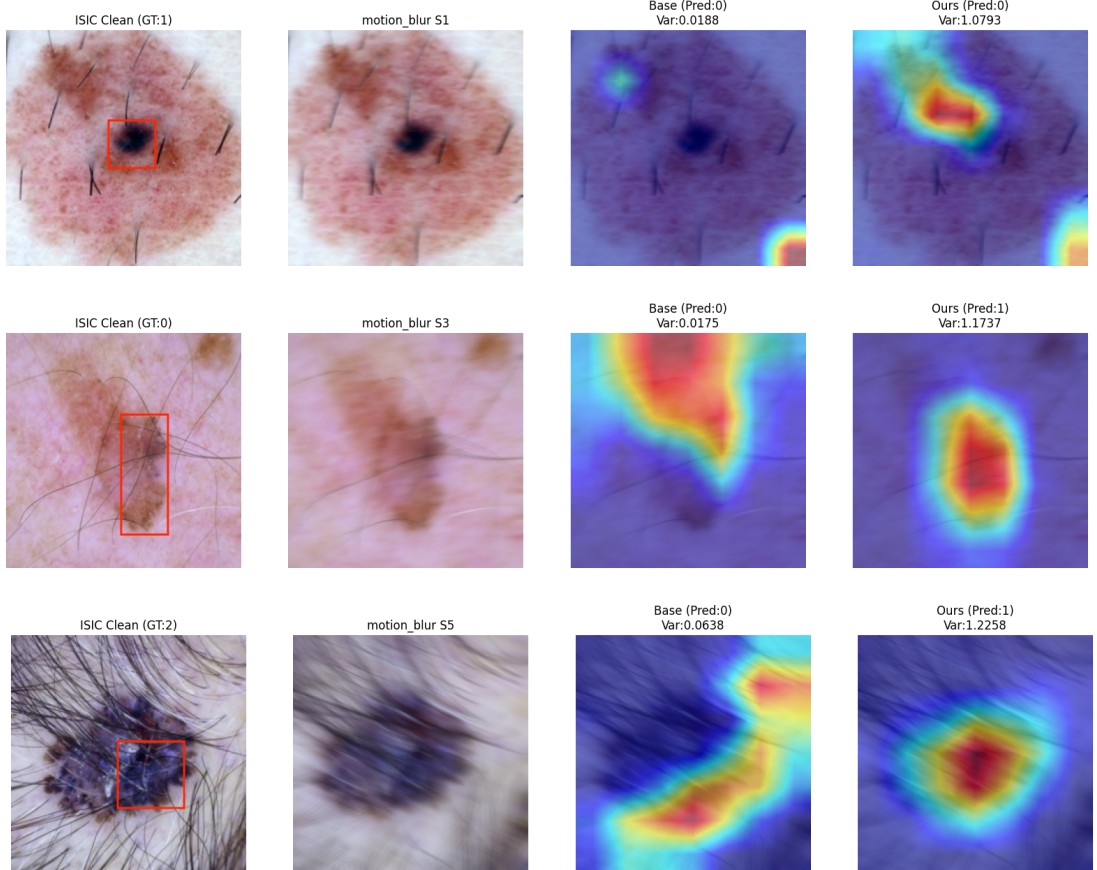

Figure 10: Qualitative comparison of failure modes for HAM10000 under motion blur. **Left to Right:** Clean image (GT), Noisy image, Baseline Grad-CAM, and Our Simplex-Aligned Grad-CAM. Even when both models output incorrect labels, our method maintains a high discriminative variance (Var) and consistently localizes the lesion area, whereas the baseline attention becomes chaotic or collapses to the majority class.

## Appendix I. Statistical Stability over Multiple Runs

To ensure the statistical significance of our findings, we conducted three independent training and evaluation runs from scratch using different random seeds (e.g., 42, 123, 999). All results reported in this section represent the **Mean ± Standard Deviation** over these runs.

### I.1. Main Performance Stability

As shown in Table 9, our Simplex-Aligned Diffusion consistently outperforms the primary baseline (DiffMIC-v2) across both benchmarks. The small standard deviations across multi-

Table 8: Sensitivity analysis of hyperparameters $\lambda$ and $\epsilon$ on HAM10000 and APTOS2019 datasets (Accuracy).

| Parameter | Dataset | Value 1 | Optimal Value | Value 2 |
|---|---|---|---|---|
| Scaling ($\lambda$) | HAM10000 | 0.8761 ($\lambda = 1.0$) | **0.8940 ($\lambda = 1.5$)** | 0.8864 ($\lambda = 2.0$) |
| | APTOS2019 | 0.8178 ($\lambda = 1.0$) | **0.8480 ($\lambda = 1.5$)** | 0.8424 ($\lambda = 2.0$) |
| Smoothing ($\epsilon$) | HAM10000 | 0.8818 ($10^{-2}$) | **0.8940 ($10^{-3}$)** | 0.8732 ($10^{-4}$) |
| | APTOS2019 | 0.8333 ($10^{-2}$) | **0.8480 ($10^{-3}$)** | 0.8443 ($10^{-4}$) |

ple runs confirm that our logit-space geometric regularization provides a stable optimization landscape, avoiding the sensitivity often associated with generative classifiers.

Table 9: Main performance comparison averaged over three independent runs (Mean $\pm$ Std).

| Dataset | Metric | DiffMIC-v2 (Baseline) | Ours (Simplex-Aligned) |
|---|---|---|---|
| HAM10000 | Accuracy ↑ | $0.8830 \pm 0.0045$ | $\mathbf{0.8932 \pm 0.0051}$ |
| | F1-Score ↑ | $0.8233 \pm 0.0101$ | $\mathbf{0.8256 \pm 0.0124}$ |
| APTOS2019 | Accuracy ↑ | $0.8385 \pm 0.0032$ | $\mathbf{0.8476 \pm 0.0025}$ |
| | F1-Score ↑ | $\mathbf{0.6687 \pm 0.0028}$ | $0.6656 \pm 0.0064$ |

## I.2. Robustness Stability and Failure Mode Analysis

We further evaluate the stability of our model under clinical artifacts. Tables 10 and 11 provide a detailed breakdown of performance degradation dynamics.

**Resistance to Mode Collapse.** A critical observation is the baseline's susceptibility to **mode collapse** under severe noise. For instance, in Table 11 under Shot Noise (Severity 5), the baseline's Cohen's Kappa remains near zero ($0.081 \pm 0.088$), indicating that its accuracy is largely driven by random guessing or majority-class prediction. In contrast, our method maintains a consistently higher and positive Kappa ($0.194 \pm 0.233$), demonstrating persistent discriminative power.

**Analysis of Shot Noise Instability.** Shot Noise exhibits the highest variance among all tested corruptions, reflecting the stochastic nature of Poisson noise in low-light sensors. Despite this inherent instability, our Simplex-Aligned strategy maintains a strictly superior mean accuracy across all severities (e.g., $0.527 > 0.375$ at S1; $0.447 > 0.390$ at S3; $0.433 > 0.243$ at S5). Our method exhibits graceful, monotonic degradation in expected performance, unlike the volatile fluctuations observed in the baseline.

Table 10: HAM10000 robustness evaluation (Mean ± Std over 3 Runs). Our method exhibits superior metric stability and significantly lower calibration error.

| Noise Type | Sev. | Method | Accuracy ↑ | Kappa ↑ | ECE ↓ |
|---|---|---|---|---|---|
| Defocus | 1 | Base | 0.748 ± 0.009 | 0.369 ± 0.041 | 0.105 ± 0.025 |
| | | Ours | **0.788 ± 0.008** | **0.518 ± 0.024** | **0.086 ± 0.023** |
| | 3 | Base | 0.674 ± 0.018 | 0.083 ± 0.041 | 0.186 ± 0.047 |
| | | Ours | **0.713 ± 0.009** | **0.279 ± 0.011** | **0.069 ± 0.028** |
| | 5 | Base | 0.655 ± 0.040 | 0.049 ± 0.134 | 0.218 ± 0.051 |
| | | Ours | **0.685 ± 0.003** | **0.238 ± 0.058** | **0.109 ± 0.019** |
| Motion | 1 | Base | 0.837 ± 0.005 | 0.632 ± 0.023 | **0.067 ± 0.029** |
| | | Ours | 0.837 ± 0.010 | **0.659 ± 0.021** | 0.098 ± 0.022 |
| | 3 | Base | 0.718 ± 0.012 | 0.280 ± 0.037 | 0.122 ± 0.031 |
| | | Ours | **0.742 ± 0.007** | **0.389 ± 0.023** | **0.062 ± 0.017** |
| | 5 | Base | 0.681 ± 0.018 | 0.165 ± 0.041 | 0.161 ± 0.049 |
| | | Ours | **0.699 ± 0.004** | **0.284 ± 0.045** | **0.090 ± 0.023** |

Table 11: APTOS2019 robustness evaluation (Mean ± Std over 3 Runs). Note the robust discriminative power (Kappa) of our method even in extreme noise regimes.

| Noise Type | Sev. | Method | Accuracy ↑ | Kappa ↑ | ECE ↓ |
|---|---|---|---|---|---|
| Shot | 1 | Base | 0.375 ± 0.204 | **0.295 ± 0.179** | 0.287 ± 0.105 |
| | | Ours | **0.527 ± 0.018** | 0.229 ± 0.150 | **0.220 ± 0.033** |
| | 3 | Base | 0.390 ± 0.187 | 0.104 ± 0.195 | 0.225 ± 0.143 |
| | | Ours | **0.447 ± 0.129** | **0.142 ± 0.133** | **0.203 ± 0.149** |
| | 5 | Base | 0.243 ± 0.167 | 0.081 ± 0.088 | **0.352 ± 0.247** |
| | | Ours | **0.433 ± 0.220** | **0.194 ± 0.233** | 0.410 ± 0.273 |
| Motion | 1 | Base | 0.731 ± 0.083 | 0.774 ± 0.126 | 0.108 ± 0.035 |
| | | Ours | **0.807 ± 0.007** | **0.855 ± 0.007** | **0.067 ± 0.008** |
| | 3 | Base | 0.589 ± 0.035 | **0.643 ± 0.068** | 0.177 ± 0.043 |
| | | Ours | **0.665 ± 0.031** | 0.638 ± 0.035 | **0.118 ± 0.009** |
| | 5 | Base | 0.512 ± 0.040 | 0.201 ± 0.038 | 0.463 ± 0.082 |
| | | Ours | **0.555 ± 0.060** | **0.424 ± 0.064** | **0.186 ± 0.053** |

