# OpenReview forum: "Simplex-Aligned Diffusion with Cross-Granularity Interaction for Robust Medical Image Classification"
_MIDL.io/2026/Conference — MIDL 2026 Poster_

### Official Review · Reviewer_sXbw · 2025-12-28

**Confidence:** 4
**Preliminary Rating:** 3
**Final Rating:** 4

**Summary:**

The authors propose Simplex-Aligned Diffusion, a generative classification framework to address the geometric mismatch in diffusion-based label generation by performing Gaussian diffusion in an unconstrained logit manifold instead of directly on one-hot labels constrained to the probability simplex. To complement the logit-space diffusion, this work introduces a Transformer-based Cross-Granularity Interaction module that fuses global and local features to stabilize semantic guidance during diffusion. Experiments on two medical image classification benchmarks demonstrate the high accuracy on clean data and improvements in robustness and calibration under corruptions.

**Strengths:**

- The work focuses on an important problem of robust and well-calibrated medical image classification, which is critical for safe clinical deployment.
- This paper includes a clear formalization of a geometric inconsistency in prior diffusion-based classifiers that operate directly on one-hot labels, along with the theoretical analysis in Appendix F, which further provides concrete intuition for the reason of probability leakage and bias arising under Gaussian diffusion.
- The proposed solution of performing diffusion in logit space via a centered log-ratio transform is mathematically principled and aligns well with known results in compositional data analysis. The Cross-Granularity Interaction module is also well-motivated.
- Empirically, the robustness and calibration evaluations are extensive.

**Weaknesses:**

- The methodological novelty is incremental, with the framework largely building upon existing diffusion-based generative classifiers and dual-stream architectures, and the main modification is the reformulation of the label space.
- The statement on application scope is somewhat overstated: the experiments are limited to two datasets and a single primary diffusion-based baseline (DiffMIC-v2) for robustness analysis. It is quite unclear whether the observed gains generalize across a broader range of medical classification settings or alternative generative classifiers. T
- The improvements on calibration are evaluated with straightforward metrics like ECE, yet the authors did not explore downstream decision-level impacts, for example, how this improved calibration would affect clinical thresholds or help clinical decision-making.

**Detailed Comments:**

- Adding a brief intuitive summary in the main text (from Appendix F) could help readers who are less familiar with diffusion models.
- Adding a comparison of inference-time cost and wall-clock runtime would be helpful.
- Some visualizations of failure cases where the method still struggles would improve transparency and help readers to better understand the limitations of this approach.

**Justification Of Final Rating:**

Thank you for your response and the additional experimental results and discussions.

Most of my concerns have been addressed. I believe this work will be of general interest to the MIDL audience, and I have raises my score to acceptance.

**Justification Of The Preliminary Rating:**

The paper presents a well-motivated contribution with theoretical grounding and robustness evaluation, particularly for calibration under distribution shift. However, the overall novelty is moderate, and the scope of empirical validation remains somewhat narrow. Open questions remain regarding scalability, inference efficiency, and downstream clinical impact. Addressing these points could strengthen the case for acceptance. In its current form, I view this work as a borderline paper between incremental improvement and a sufficiently general methodological advance.

**Questions To Address In The Rebuttal:**

- How does the proposed Simplex-Aligned Diffusion generalize to multi-class problems with a much larger number of categories, where the logit dimensionality increases substantially?
- Can the authors add runtime or latency comparisons against discriminative baselines to clarify the practical trade-offs of diffusion-based inference?
- To what extent can the calibration improvements be translated into clinically actionable benefits?
- Have the authors tested the method on datasets beyond the current scope, or can you comment on expected behavior in other modalities?

---

> ### Author Response · Authors · 2026-01-25
>
> We thank the reviewer for the insightful comments. We have strictly addressed each point and updated the manuscript accordingly.
>
> **Q1.Scalability to multi-class problems and logit dimensionality**
>
> We clarify that the computational complexity of Simplex-Aligned Diffusion does not scale substantially with the number of categories ($K$). While our logit vector $z_0$ is length $K$, it resides on a $(K-1)$-dimensional manifold due to the sum-to-zero constraint of the Centered Log-Ratio (CLR) transform. Even for 1,000 categories, a 999-D intrinsic manifold is orders of magnitude smaller than the pixel manifold (e.g., $\approx 150,000$ dimensions for a $224\times224$ image). Thus, the denoising overhead remains mathematically and computationally negligible compared to the visual feature extraction.
>
> **Q2.Brief intuitive summary in the main text**
>
> Following your suggestion, we have added a brief, intuitive summary of the theoretical analysis from Appendix F into Section 3.1 of the main text. This summary explains how the mismatch between Gaussian noise and the simplex boundary leads to "probability leakage" and systematic bias, providing immediate intuition for readers without requiring them to refer to the full proof in the Appendix.
>
> **Q3. Latency comparisons against baselines**
>
> We have added a comprehensive efficiency analysis in Table R3. Our method reduces GFLOPs by 39% compared to the SOTA diffusion baseline (DiffMIC-v2).
>
> **Table R3: Computational Efficiency and Hardware Metrics**
>
> | Metric | ResNet50 | DiffMIC-v2 (Baseline) | Simplex-Aligned (Ours) |
> | :--- | :---: | :---: | :---: |
> | Inference Latency | 0.36 ms/img | 2.51 ms/img | 2.40 ms/img |
> | Total GFLOPs | 2.05 G | 277.99 G | 168.80 G |
> | Peak GPU Memory | ~0.90 GB | 2.37 GB | 1.95 GB |
> | Throughput | ~2700+ FPS | ~398 FPS | ~416 FPS |
>
> While our method has higher theoretical FLOPs than standard CNNs, FLOPs alone are not a reliable proxy for inference latency in diffusion-based classifiers. Unlike CNNs that operate on high-resolution pixel space in a single forward pass, our method performs denoising in a low-dimensional logit space, where the iterative steps are highly optimized and GPU-parallelizable. As shown in Table R3, this results in competitive wall-clock latency (2.40 ms/img), moderate memory usage (1.95 GB), and practical throughput, despite a higher FLOP count.
>
> Importantly, the reported 3×H100 configuration was used solely to accelerate training and does not reflect inference-time requirements. In practice, inference and deployment require only a single GPU with modest memory. We emphasize that our goal is not to match the raw efficiency of standard CNNs, but to substantially reduce the computational overhead of diffusion-based classifiers while achieving improved robustness and calibration.
>
>
>
>
> **Q4. Clinical actionability of calibration improvements**
>
> The primary clinical benefit of lower ECE is decision reliability. By ensuring that the model’s confidence score accurately reflects its predictive accuracy, clinicians can implement safer automated triaging. High-confidence predictions become truly trustworthy, reducing the risk of "overconfident misdiagnosis" that typically plagues miscalibrated discriminative models.
>
> **Q5.Visualizations of failure cases (Appendix G)**
>
> We have added a new Appendix G providing a visual analysis of failure modes under extreme corruptions (Shot Noise and Motion Blur).
>
> Explainable Failure: The Grad-CAM heatmaps show that even when our model fails to identify the correct class under severe noise, it consistently localizes the lesion area. In contrast, the baseline (DiffMIC-v2) heatmaps under the same conditions become sparse or chaotic, losing all semantic focus. This demonstrates that our method maintains a "meaningful failure" mode, providing persistent visual guidance even when the diagnostic signal is degraded.
>
> **Q6. Generalization to other modalities**
>
> The Simplex-Aligned formulation addresses a geometric inconsistency that arises in diffusion-based labeling. While our current experiments focus on 2D medical images, the underlying principles are not specific to a particular modality. We have added a discussion in the Conclusion noting the potential extensibility of our framework to 3D volumes (e.g., CT/MRI) and 1D signals, where similar logit-space geometric regularization principles may be applicable to improve robustness.
>
> Overall, our contribution is intentionally focused on a targeted geometric refinement of the label diffusion process, rather than introducing additional architectural complexity.

---

> ### Comment · Area_Chair_sp77 · 2026-01-31
> **Please provide final rating**
>
> Thank you for the detailed review. Please provide your final rating based on the authors’ response by clicking ‘Edit’ → ‘Official Review’. Thank you.

---

### Official Review · Reviewer_swap · 2026-01-07

**Confidence:** 3
**Preliminary Rating:** 4
**Final Rating:** 4

**Summary:**

This paper focuses on generative classifier approaches in medical imaging. Recent SOTA methods like DiffMIC-v2 suffer from a critical flaw in that they apply unbounded Gaussian noise directly to bounded, discrete one-hot label simplices. This mismatch forces prediction into invalid probability spaces leading to model unreliability and overconfidence (due to underestimation of probability values). This is a major issue in medical imaging classification systems which relies on trustworthiness including providing calibrated uncertainty estimates and maintain robustness under acquisition distribution shifts.

To deal with this problem, the authors propose Simplex-aligned Diffusion which maps the probability simplex to a Euclidean space thereby ensuring mathematical consistency with Gaussian diffusion. A Transformer-based cross granularity interaction module is also introduced to dynamically model global-local dependencies and stabilize visual guidance. Experiments are done on the APTOS2019 (Diabetic Retinopathy) and HAM10000 (Skin Lesion) datasets.

**Strengths:**

1. The paper is exceptionally well-written and has a solid theoretical foundation.

2. While the concept of implementing diffusion on the simplex via logit-normal transformations and using discrete diffusion kernels is well-established, the application to medical imaging classification is indeed novel.

3. The results, especially those quantifying robustness in the presence of simulated sensor noise and handheld motion blur, show an improvement in calibration/robustness and reliability over previous methods. More importantly, the proposed method does not collapse (calculating accuracy by focusing on only the majority label), which is important from a clinical safety standpoint.

4. The transformer-based tokenizer which models dependencies between the global and local patch tokens produces a fusion prior and refined semantic features versus using simple concatenation (in the baselines). This helps in conditioning the diffusion process.

**Weaknesses:**

There are no major weaknesses. However, I would like to raise a few discussion points which the authors can comment on:

1. The limitation of high iterative sampling cost is highlighted in the conclusion section. However, nowhere in the paper is this quantified. It would be great to get more details on this compared to methods like DiffMIC-v2. Example: how many timesteps per image are required for classification and how computationally expensive is it vs inference using a standard CNN ?

2. Given the computational costs involved, how would this method compare against, say, uncertainty-aware discriminative models to achieve similar levels of robustness and calibration minus the computational costs ?

3. Why has the evaluation been done only on 2D datasets versus using 3D datasets (Ct, MRI). Using the latter would have been an interesting test for the Transformer-based tokenizer to see if the cross-granularity module works well with 3D volumes showing long-range dependencies.

**Detailed Comments:**

Please refer to the comments in the above sections. No further comments have been added here.

**Justification Of Final Rating:**

The authors have satisfactorily addressed the review comments and made changes to the paper that make it suitable for publication to the MIDL conference. After careful deliberation, I have decided to stick to my original rating of 4.

**Justification Of The Preliminary Rating:**

The paper is theoretically strong and backed by rigorous experiments. The results indeed shown an improvement and this would be helpful for improving trustworthiness and safety of medical image classification systems.

**Questions To Address In The Rebuttal:**

1. Explanation of the high iterative sampling cost in performing the classification ?
2. Comparison against other method such as uncertainty-aware discriminative models ?
3. Justification for evaluation being done only on 2D datasets versus using 3D datasets (Ct, MRI).

---

> ### Author Response · Authors · 2026-01-25
>
> We thank the reviewer for the positive assessment and the recognition of our "solid theoretical foundation" and "novel application." We particularly appreciate the insightful questions regarding computational costs and the scope of our evaluation. Below are our point-by-point responses.
>
> **Q1. Explanation of the high iterative sampling cost in performing the classification.**
>
> We appreciate the reviewer’s request for a quantitative characterization of computational cost. While diffusion-based classifiers involve iterative sampling, our Simplex-aligned Diffusion is substantially more efficient than prior reconstruction-based diffusion classifiers.
>
> Our method operates on a low-dimensional logit manifold rather than the high-dimensional pixel space, which significantly reduces per-step computation.
> As summarized in Table R3, our method reduces the total computational burden by approximately 39% GFLOPs compared to DiffMIC-v2 (168.8G vs. 277.99G), while achieving higher throughput (416 FPS vs. 398 FPS) and slightly lower inference latency (2.40 ms/img vs. 2.51 ms/img).
>
>
> **Table R3: Computational Efficiency and Hardware Metrics**
>
> | Metric | ResNet50 | DiffMIC-v2 (Baseline) | Simplex-Aligned (Ours) |
> | :--- | :---: | :---: | :---: |
> | Inference Latency | 0.36 ms/img | 2.51 ms/img | 2.40 ms/img |
> | Total GFLOPs | 2.05 G | 277.99 G | 168.80 G |
> | Peak GPU Memory | ~0.90 GB | 2.37 GB | 1.95 GB |
> | Throughput | ~2700+ FPS | ~398 FPS | ~416 FPS |
>
>
> While standard CNNs remain faster, our method’s peak memory (1.95 GB) makes it highly accessible for deployment on standard clinical workstations. The iterative cost is a trade-off for the superior robustness and calibration required for safety-critical medical decisions.
>
> **Q2. Comparison against uncertainty-aware discriminative models.**
>
> We agree that uncertainty-aware discriminative models (e.g., MC-Dropout, Deep Ensembles, evidential learning, and post-hoc calibration) can be significantly faster and are strong alternatives. A key practical difference is that these methods typically improve uncertainty either by approximating a predictive distribution via repeated forward passes (dropout/ensembles) or by introducing additional calibration objectives/procedures, and their reliability under acquisition shifts can still be sensitive to how the shift manifests and how calibration is performed.
>
> Our method is complementary: it is a generative classifier and directly models class-conditional structure. More importantly, we address a specific failure mode in diffusion-based generative classifiers: unbounded Gaussian noise injected into bounded simplex variables can create a geometric mismatch, leading to unreliable probabilities and overconfidence. By performing diffusion in a simplex-aligned space, we obtain consistently improved robustness and calibration under the evaluated corruptions at a practical inference cost (2.40 ms/img). Thus, rather than aiming to match the raw speed of a standard CNN, our goal is to deliver more trustworthy predictions under clinically relevant acquisition shifts with substantially lower overhead than reconstruction-based diffusion classifiers.
>
> **Q3. Justification for evaluation on 2D datasets versus 3D datasets**
>
> We focused on 2D datasets (APTOS2019 and HAM10000) for two primary reasons. First, these are the standard benchmarks used by the most recent SOTA methods (including DiffMIC-v2), allowing for a direct and fair comparison of the Simplex-aligned theory. Second is for theoretical validation as our primary goal was to solve the mathematical mismatch between Gaussian noise and bounded simplices. 2D imaging provides a high-clarity environment to validate this alignment without the confounding architectural complexities of 3D volumetric processing. We agree that the Transformer-based cross-granularity module is uniquely suited for 3D volumes (e.g., modeling long-range dependencies across slices). We consider the extension to 3D modalities like CT and MRI a high-priority direction for our future work.

---

> ### Comment · Area_Chair_sp77 · 2026-01-31
> **Please provide final rating**
>
> Thank you for the detailed review. Please provide your final rating based on the authors’ response by clicking ‘Edit’ → ‘Official Review’. Thank you.

---

### Official Review · Reviewer_Y67o · 2026-01-09

**Confidence:** 4
**Preliminary Rating:** 4

**Summary:**

The authors propose Simplex-Aligned Diffusion, a novel framework for generative medical image classification that addresses a geometric conflict in existing diffusion-based classifiers (e.g., DiffMIC). Standard methods apply unbounded Gaussian noise to bounded one-hot label vectors, which the authors identify as a source of bias and instability. To resolve this, the proposed method reformulates the label generation process on the continuous logit manifold using a Center Log-Ratio (CLR) transformation, ensuring consistency with Gaussian diffusion dynamics. Additionally, a Cross-Granularity Interaction module based on a Transformer encoder is introduced to dynamically model dependencies between global image contexts and local lesion features. Extensive experiments on the APTOS2019 and HAM10000 datasets demonstrate that the method achieves competitive accuracy on clean data while outperforming state-of-the-art baselines in uncertainty calibration (ECE) and robustness against clinical artifacts like sensor noise.

**Strengths:**

Theoretical motivation: The paper tackles a theoretical flaw in current generative classifiers: the mismatch between the bounded probability simplex and the unbounded Gaussian noise used in diffusion models. The theoretical analysis provided in Appendix F (Probability Leakage and Approximation Gap) is rigorous and convincingly explains why prior methods suffer from overconfidence. Shifting the diffusion process from the discrete/simplex space to the unconstrained logit space via the CLR transformation is a mathematically elegant solution. While logit-space diffusion exists in general generative modeling, its application to discriminative medical tasks to enforce geometric consistency is a valuable contribution.

Robustness results: The empirical results, particularly the Continuous Stress Test (Figure 2), are remarkable. The method avoids the "mode collapse" observed in the baseline (DiffMIC-v2) under high noise, maintaining a reasonable Cohen's Kappa where the baseline drops to near zero.

Evaluation: The authors go beyond standard Gaussian noise and evaluate strictly domain-specific artifacts (Shot Noise for fundus, Motion Blur for dermatoscopy).

**Weaknesses:**

Lack of Error Estimates: The paper reports single-point metric estimates (e.g., "0.894") without standard deviations or confidence intervals. Deep learning training is stochastic, and without error bars derived from multiple independent runs (e.g., N=3 or 5), it is difficult to determine if marginal improvements (e.g., 1.1% accuracy gain on HAM10000) are statistically significant or due to random seed variance.

Counter-intuitive Performance Trends (U-Shape): In Table 2, under Shot Noise for APTOS2019, the proposed method exhibits a puzzling non-monotonic trend. Accuracy drops from 0.507 (Severity 1) to ~0.299 (Severity 3), but then sharply rises to ~0.608 at Severity 5. Cohen's Kappa follows this same trend (improving at high noise), suggesting this is not simple mode collapse. This behavior contradicts the expectation that information content decreases as noise increases and requires explanation. Could this be due to chance (training stochasticity), or does this specific noise addition inadvertently highlight traits that the classifier uses?

Efficiency: The authors identify computational cost as a barrier to clinical practice, pointing out that reconstruction-based diffusion methods are "computationally prohibitive". However, the implementation details reveal that experiments required 3 NVIDIA H100 GPUs. This high-end hardware requirement implies a similar "computationally prohibitive cost" for clinical deployment compared to standard CNNs. Additionally, it would have been interesting to see the inference time of the autohr's method vs. the DiffMIC-v2 baseline.

**Detailed Comments:**

Caption Brevity: Some table and figure captions are too brief and do not self-contain the main takeaway. For instance, the caption for Table 3 is simply "Ablation Study on Component Contributions", and Figure 3 is "Qualitative comparison of attention maps (Grad-CAM)". Expanding these would improve readability.

Clarification on Loss: In Equation (4), the loss is weighted by $\omega(v_{trans})$. The text mentions this is a "focal term," but the exact functional form of $\omega$ is not defined in the main text.

**Justification Of The Preliminary Rating:**

The paper addresses a distinct and mathematically demonstrable problem in generative medical classification (the geometric mismatch in diffusion). The solution proposed (Simplex-Aligned Diffusion) is elegant, theoretically grounded, and yields substantial empirical gains in robustness and calibration. However, the evaluation rigor has gaps that prevent a higher rating: specifically the lack of error bars for statistical significance and the unexplained U-shaped performance anomaly in the robustness benchmarks. If the authors can satisfactorily address the U-shape and provide confidence intervals, I would be willing to raise my score.

**Questions To Address In The Rebuttal:**

Performance U-Shape: In Table 2 (APTOS / Shot Noise), your method's accuracy and Kappa drop at Severity 3 but significantly recover at Severity 5 (outperforming Severity 1). Can you explain this counter-intuitive behavior? Is it a stochastic anomaly, an artifact of the specific noise type, or a table error?

Statistical Significance: Can you provide the mean and standard deviation for your main results (Table 1 and Table 2) over multiple training runs? (If this is not possible due to time constraints, please clarify and remove the "significantly better" claims).

Sensitivity Analysis: How sensitive is the model to the scaling factor $\lambda$ and the label smoothing $\epsilon$? Did you experiment with learnable scaling?

---

> ### Author Response · Authors · 2026-01-25
> **Response Part 1**
>
> We thank the reviewer for their insightful feedback and for recognizing the theoretical elegance and empirical gains of our Simplex-Aligned Diffusion strategy. We have addressed all concerns regarding statistical rigor, performance trends, and computational efficiency through comprehensive multi-seed evaluations and technical analysis.
>
> **Q1. Statistical Significance (Lack of Error Estimates)**
>
> We agree that error estimates are essential for validating performance gains. We conducted three independent training runs with different random seeds for both our method and the primary baseline. Each run was fully trained from scratch, and all reported results are obtained by averaging the test-time performance of the independently trained models.
>
> **Table R1: Main Performance Comparison (Mean ± Std over 3 Runs)**
>
>
> | Dataset | Metric | DiffMIC-v2 (Baseline) | Ours (Simplex-Aligned) |
> | :--- | :--- | :--- | :--- |
> | HAM10000 | Accuracy | 0.8830 ± 0.0045 | 0.8932 ± 0.0051 |
> | | F1-Score | 0.8233 ± 0.0101 | 0.8256 ± 0.0124 |
> | APTOS2019 | Accuracy | 0.8385 ± 0.0032 | 0.8476 ± 0.0025 |
> | | F1-Score | 0.6687 ± 0.0028 | 0.6656 ± 0.0064 |
>
> As shown in Table R1, our method consistently achieves higher Accuracy with small standard deviations, confirming the stability of our strategy.
>
> **Q2. Counter-intuitive "U-Shape" Performance Trend**
>
> Regarding the counter-intuitive “U-shape” observed under Shot Noise on APTOS2019 (Table 2), we confirm that this behavior is a single-seed stochastic outlier, rather than a stable or reproducible property of our method.
>
>
> **Table R2(a): HAM10000 Robustness Evaluation (Mean ± Std over 3 Runs)**
>
> | Noise Type | Sev. | Accuracy (Base / Ours) | Kappa (Base / Ours) | ECE (Base / Ours) |
> | :--- | :---: | :---: | :---: | :---: |
> | Defocus | 1 | 0.748±0.009 / 0.788±0.008 | 0.369±0.041 / 0.518±0.024 | 0.105±0.025 / 0.086±0.023 |
> | Defocus | 3 | 0.674±0.018 / 0.713±0.009 | 0.083±0.041 / 0.279±0.011 | 0.186±0.047 / 0.069±0.028 |
> | Defocus | 5 | 0.655±0.040 / 0.685±0.003 | 0.049±0.134 / 0.238±0.058 | 0.218±0.051 / 0.109±0.019 |
> | Motion | 1 | 0.837±0.005 / 0.837±0.010 | 0.632±0.023 / 0.659±0.021 | 0.067±0.029 / 0.098±0.022 |
> | Motion | 3 | 0.718±0.012 / 0.742±0.007 | 0.280±0.037 / 0.389±0.023 | 0.122±0.031 / 0.062±0.017 |
> | Motion | 5 | 0.681±0.018 / 0.699±0.004 | 0.165±0.041 / 0.284±0.045 | 0.161±0.049 / 0.090±0.023 |
>
> **Table R2(b): APTOS2019 Robustness Evaluation (Mean ± Std over 3 Runs)**
> | Noise Type | Sev. | Accuracy (Base / Ours) | Kappa (Base / Ours) | ECE (Base / Ours) |
> | :--- | :---: | :---: | :---: | :---: |
> | Shot | 1 | 0.375±0.204 / 0.527±0.018 | 0.295±0.179 / 0.229±0.150 | 0.287±0.105 / 0.220±0.033 |
> | Shot | 3 | 0.390±0.187 / 0.447±0.129 | 0.104±0.195 / 0.142±0.133 | 0.225±0.143 / 0.203±0.149 |
> | Shot | 5 | 0.243±0.167 / 0.433±0.220 | 0.081±0.088 / 0.194±0.233 | 0.352±0.247 / 0.410±0.273 |
> | Motion | 1 | 0.731±0.083 / 0.807±0.007 | 0.774±0.126 / 0.855±0.007 | 0.108±0.035 / 0.067±0.008 |
> | Motion | 3 | 0.589±0.035 / 0.665±0.031 | 0.643±0.068 / 0.638±0.035 | 0.177±0.043 / 0.118±0.009 |
> | Motion | 5 | 0.512±0.040 / 0.555±0.060 | 0.201±0.038 / 0.424±0.064 | 0.463±0.082 / 0.186±0.053 |
>
> To assess reproducibility, we conducted three independent train-and-evaluation runs with different random seeds and report Mean ± Std in Table R2. Under multi-seed averaging, the apparent “U-shape” observed in the original single-seed results disappears, and performance degrades monotonically in expectation.
>
> For Shot Noise on APTOS2019, mean Accuracy decreases as $0.527 \to 0.447 \to 0.433$, with no systematic recovery in Cohen’s Kappa at high noise. Notably, the standard deviation at S5 is comparable to the mean, indicating a high-variance regime in which single-seed evaluations are susceptible to outlier fluctuations. Consistent with this, additional seeds (e.g., 123 and 999) do not exhibit stable high-noise recovery, and the previously observed S5 spike (seed = 42) falls within the Mean ± Std range in Table R2.
>
> This non-monotonic behavior is largely confined to Shot Noise. In contrast, Motion Blur and Defocus Blur exhibit stable multi-seed degradation, which we attribute to their spatially smooth, convolutional nature that preserves global structure. Shot Noise, arising from stochastic photon counting at the pixel level, introduces discrete and uncorrelated perturbations, leading to higher variance across seeds.
>
> Importantly, this behavior is not indicative of mode collapse: under multi-seed averaging, our method maintains positive and higher Cohen’s Kappa at moderate and high noise levels (S3/S5) and consistently outperforms the baseline in mean Accuracy across all Shot Noise severities.

---

> > ### Author Response · Authors · 2026-01-25
> > **Response Part 2**
> >
> > **Q3. Efficiency and Deployment Costs**
> >
> > We have performed a head-to-head comparison under identical hardware configurations.
> >
> > **Table R3: Computational Efficiency and Hardware Metrics**
> >
> > | Metric | ResNet50 | DiffMIC-v2 (Baseline) | Simplex-Aligned (Ours) |
> > | :--- | :---: | :---: | :---: |
> > | Inference Latency | 0.36 ms/img | 2.51 ms/img | 2.40 ms/img |
> > | Total GFLOPs | 2.05 G | 277.99 G | 168.80 G |
> > | Peak GPU Memory | ~0.90 GB | 2.37 GB | 1.95 GB |
> > | Throughput | ~2700+ FPS | ~398 FPS | ~416 FPS |
> >
> > While our method has higher theoretical FLOPs than standard CNNs, FLOPs alone are not a reliable proxy for inference latency in diffusion-based classifiers. Unlike CNNs that operate on high-resolution pixel space in a single forward pass, our method performs denoising in a low-dimensional logit space, where the iterative steps are highly optimized and GPU-parallelizable. As shown in Table R3, this results in competitive wall-clock latency (2.40 ms/img), moderate memory usage (1.95 GB), and practical throughput, despite a higher FLOP count.Notably, our peak memory requirement of only 1.95 GB confirms that the model is easily deployable on standard consumer-grade hardware, such as an RTX 4090 (24GB), facilitating practical clinical integration.
> >
> > Importantly, the reported 3×H100 configuration was used solely to accelerate training and does not reflect inference-time requirements. In practice, inference and deployment require only a single GPU with modest memory. We emphasize that our goal is not to match the raw efficiency of standard CNNs, but to substantially reduce the computational overhead of diffusion-based classifiers while achieving improved robustness and calibration.
> >
> >
> >
> > **Q4. Sensitivity Analysis ($\lambda$ and $\epsilon$)**
> >
> > We conducted extensive ablation studies to investigate the impact of the scaling factor $\lambda$ and label smoothing $\epsilon$.
> >
> > **Table R4: Sensitivity to Hyperparameters (Accuracy)**
> >
> > | Parameter | Dataset | Value 1 | Optimal Value | Value 2 |
> > | :--- | :---: | :---: | :---: | :---: |
> > | Scaling (λ) | HAM10000 | 0.8761 (λ=1.0) | 0.8940 (λ=1.5) | 0.8864 (λ=2.0) |
> > | Scaling (λ) | APTOS2019 | 0.8178 (λ=1.0) | 0.8480 (λ=1.5) | 0.8424 (λ=2.0) |
> > | Smoothing (ε) | HAM10000 | 0.8818 (1e−2) | 0.8940 (1e−3) | 0.8732 (1e−4) |
> > | Smoothing (ε) | APTOS2019 | 0.8333 (1e−2) | 0.8480 (1e−3) | 0.8443 (1e−4) |
> >
> >
> > **Scaling factor λ.**
> > As shown in Table R4, model performance exhibits a clear single-peak behavior with respect to the scaling factor λ, achieving the best results at λ = 1.5 on both HAM10000 and APTOS2019. When moving away from this value, performance degrades smoothly rather than collapsing abruptly. Smaller λ under-utilizes the simplex alignment, while excessively large λ leads to more over-confident predictions, which slightly harms robustness on ambiguous samples. Overall, the observed trend indicates that the method is not overly sensitive to λ, and a fixed λ = 1.5 provides a stable and reproducible operating point.
> >
> > **Label smoothing ε.**
> > For label smoothing, ε = 10⁻³ consistently yields the best accuracy across both datasets. Larger ε (10⁻²) overly softens the label distribution and reduces discriminative power, whereas smaller ε (10⁻⁴) approaches hard one-hot labels and weakens the numerical stability of the CLR transformation. Importantly, performance varies smoothly across the tested ε values, suggesting that the model remains robust within a reasonable range of smoothing strengths.
> >
> > **Learnable scaling.**
> > We considered making λ a learnable parameter, but opted for a fixed global scaling in our final design. Since λ directly controls the geometry of the simplex-aligned logit transformation, learning it jointly with the model can introduce additional optimization instability and reduce interpretability. Given the smooth and well-behaved sensitivity trends observed in Table R4, a fixed λ offers a favorable balance between performance, stability, and ease of deployment.
> >
> > **Q5. Technical Clarifications**
> >
> > The focal-style weighting term in Eq. (4) is now explicitly and rigorously defined in **Section 3.3**, making the diffusion objective fully self-contained.
> > All table and figure captions have also been revised to clearly describe the experimental settings and key findings.

---

> ### Comment · Area_Chair_sp77 · 2026-01-31
> **Please provide final rating**
>
> Thank you for the detailed review. Please provide your final rating based on the authors’ response by clicking ‘Edit’ → ‘Official Review’. Thank you.

---

> ### Author Response · Authors · 2026-02-02
>
> Dear Reviewer Y67o,
>
> Thanks again for the detailed feedback. We’ve added the requested follow-ups: multi-seed Mean±Std for main + robustness results (Tables R1–R2), an explanation showing the APTOS Shot Noise “U-shape” is a single-seed outlier under multi-seed evaluation (Table R2(b)), and an efficiency comparison (Table R3) plus λ/ε sensitivity (Table R4) and loss definition clarifications.
>
> If you have time, we’d really appreciate any final thoughts or remaining concerns. Thank you!

---

### Author Rebuttal · Authors · 2026-01-25

**Rebuttal:**

We sincerely thank the reviewers (**Y67o, swap, sXbw**) for their constructive and insightful feedback on the *Simplex-Aligned Diffusion* framework.
Based on their comments, we have carefully categorized the primary concerns and implemented substantial revisions in the updated manuscript.

---

### I. Consolidation of Core Reviewer Concerns

- **Statistical Rigor and “U-Shape” Anomaly**
  We address the lack of error estimates by reporting multi-seed results and provide an explanation of the single-seed stochasticity that caused the previously observed non-monotonic trend in the Shot Noise benchmarks.

- **Computational Efficiency**
  We quantify inference latency, GFLOPs, and hardware overhead in comparison with both discriminative and generative baselines, in order to assess the feasibility of clinical deployment.

- **Failure Analysis and Reliability**
  We investigate *Meaningful Failure* modes using Grad-CAM visualizations and introduce discriminative variance to verify that visual guidance remains semantically grounded even under misclassification.

- **Theoretical Intuition and Generalization**
  We clarify the motivation for operating on the logit manifold for readers less familiar with diffusion models and discuss the potential extension of the framework to 3D and 1D medical modalities.

---

### II. Summary of Major Revisions

### Statistical Stability Assessment *(Appendix I)*
- All primary results in **Table 1** and **Table 2** are updated to report **Mean ± Standard Deviation** over three independent runs.

### Computational Efficiency Benchmarking *(Table 4)*
- Section **4.5** now provides a detailed comparison of **Inference Latency**, **Total GFLOPs**, **Peak GPU Memory**, and **Throughput (FPS)** across **ResNet-50**, **DiffMIC-v2**, and our method.

### Qualitative Failure Analysis *(Appendix G)*
- We provide Grad-CAM visualizations under severe clinical artifacts.
- We mathematically define **Logit Variance** as a proxy for diagnostic intent, demonstrating persistent lesion localization even when pictions are incorrect.

### Hyperparameter Sensitivity Study *(Appendix H)*
- A comprehensive ablation study on the scaling factor $\lambda$ and label smoothing parameter $\epsilon$ is conducted, confirming the robustness of the selected operating point.


All revisions are highlighted in red, and we provide a detailed point-by-point response to each reviewer.

**Supporting Material:**

/attachment/ea678e4b14215495b16783be90f7ea24116d804b.pdf

---

### Meta-Review · Area_Chair_sp77 · 2026-02-07

**Recommendation:** Accept (Poster)
**Confidence:** 4

**Metareview:**

Overall, the paper received positive evaluations. All reviewers commended the strength and clarity of the theoretical foundation. The authors were actively engaged during the discussion period. The primary concerns raised during the review process were adequately addressed through the additional experiments and clarifications provided in the rebuttal.

---

### Decision · Program_Chairs · 2026-02-13

Accept (Poster)